



# On the Robustness of the Surface Response to Austral Stratospheric Polar Vortex Extremes

Nora Bergner[1,2], Marina Friedel[1], Daniela I.V. Domeisen[1,3], Darryn Waugh[4], and Gabriel Chiodo[1]

[1]Institute of Atmospheric and Climate Science, ETH Zürich, Zurich, Switzerland
[2]Extreme Environments Research Laboratory, EPFL Valais, Sion, Switzerland
[3]Institute of Earth Surface Dynamics, University of Lausanne, Lausanne, Switzerland
[4]Department of Earth and Planetary Sciences, Johns Hopkins University, Baltimore, MD, USA

**Correspondence:** Nora Bergner (nora.bergner@epfl.ch)

**Abstract.** Extreme events in the stratospheric polar vortex can lead to changes in the tropospheric circulation and impact the surface climate on a wide range of timescales. The austral stratospheric vortex shows its largest variability in spring, and a weakened polar vortex is associated with changes in the spring to summer surface climate, including hot and dry extremes in Australia. However, the robustness and extent of the connection between polar vortex strength and surface climate on inter-
annual timescales remain unclear. We assess this relationship by using reanalysis data and simulations from two independent chemistry-climate models (CCMs), building on previous work that is mainly based on observations. The CCMs show a similar downward propagation of polar vortex anomalies as the reanalysis data and weak (strong) polar vortex anomalies are on average followed by a negative (positive) tropospheric Southern Annular Mode (SAM) in spring to summer. The signature in the surface climate following polar vortex weakenings is characterized by high surface pressure and warm temperature anomalies
over Antarctica, the region where surface signals are most robust across all model and observational datasets. However, the tropospheric SAM response in the models is inconsistent with observations. In one CCM, the SAM is more negative compared to the reanalysis after weak polar vortex events, whereas in the other CCM, it is less negative. In addition, both models do not reproduce all the regional changes in midlatitudes, such as the warm and dry anomalies over Australia. We find that these inconsistencies are linked to model biases in the basic state, such as the latitude of the eddy-driven jet and the persistence of the tropospheric SAM. Furthermore, bootstrapping of the data reveals sizable uncertainty in the magnitude of the surface
signals in both models and observations due to internal variability. Our results demonstrate that anomalies of the austral stratospheric vortex have significant impacts on surface climate, although the ability of models in capturing regional effects across the Southern Hemisphere is limited by biases in their representation of the tropospheric circulation.



# 1 Introduction

Variability in the stratospheric polar vortex can influence surface climate on timescales of weeks to months (Baldwin and Dunkerton, 1999, 2001). For example, circulation anomalies during Sudden Stratospheric Warmings (SSWs), events where upward propagating and dissipating waves rapidly decelerate the stratospheric zonal flow, can descend to the lower stratosphere and impact the tropospheric circulation. In the Northern Hemisphere (NH), where SSWs occur approximately every other year, these events are linked to surface extremes in midlatitude regions, for example cold-air outbreaks in North America and
Eurasia (Scaife et al., 2008; Kolstad et al., 2010; Domeisen and Butler, 2020). In the Southern Hemisphere (SH), the polar vortex is stronger, less variable and persists further into spring than its NH counterpart, and SSWs are very rare (2002 was the only recorded major SSW). These hemispheric differences are due to differing distributions of land surface and topography, resulting in weaker tropospheric wave disturbances in the SH (Plumb, 1989).

Despite major SSWs being extremely rare, the austral polar vortex shows some interannual variability, especially in late
winter and spring, when increased insolation, paired with stronger wave forcing, lead to a more disturbed vortex. These polar vortex weakenings are similar to minor SSWs that also occur regularly in the Northern Hemisphere, where the zonal flow is weakened, but no complete wind reversal takes place. Stratospheric polar vortex anomalies can influence the polarity of the Southern Annular mode (SAM) in the troposphere (Thompson et al., 2005), and as a result, can also affect SH surface climate. This influence is long lasting and extends to the entire late spring to summer season between October and January (Lim et al.,
2018). For example, austral polar vortex weakenings have been suggested as drivers of surface extremes in southern and eastern Australia, enhancing the probability of hot and dry extremes and fire risk with severe impacts for humans and ecosystems (Lim et al., 2019). This connection between perturbations in the winter- to springtime polar vortex and subsequent surface climate shows the potential for skillful seasonal prediction for both stratospheric and tropospheric conditions between August and February (Byrne and Shepherd, 2018; Lim et al., 2018; Domeisen et al., 2020). Enhanced predictability could improve early
adaptation to reduce the negative impacts of extreme heat and drought on people and ecosystems. These vortex weakenings have also been linked to surface climate in Antarctica, with cooling over the Antarctic peninsula and warming over the rest of Antarctica (Kwon et al., 2020), with potential knock-on effects on ice sheet mass balance.

The vast majority of previous studies on SH stratosphere-troposphere coupling on interannual timescales is based on station and reanalysis data. Even though some of the regional signals, such as the hot and dry extremes over Australia, have been
extensively studied, statistical uncertainty is large given the relatively short observational record in the SH and the small sample size of anomalous polar vortex events (depending on the definition, between 10-15 events (Lim et al., 2018, 2019; Byrne and Shepherd, 2018; Kwon et al., 2020)). The robustness and spatial extent of the downward impact of the stratosphere in the SH thus remains unclear. Another observation-based method to investigate the robustness of a surface composite to sampling variability was applied by Oehrlein et al. (2021) for the NH SSW surface impacts. This method based on bootstrapping has been
previously used to examine the extratropical response to ENSO in the NH (Deser et al., 2017). Oehrlein et al. (2021) randomly resample observed SSW events to create synthetic bootstrapped surface composites that could have plausibly occurred with a different sequence of atmospheric variability unrelated to the polar vortex. They find that the pattern of synthetic composites is





consistent with the known surface response of SSWs, but that the magnitude and spatial pattern is highly variable. They further find that the uncertainty in the SSW surface composite is largely independent from the strength of the stratospheric perturbation but results from internal tropospheric variability. Similarly, tropospheric variability may also play a role in the surface pattern observed after weak vortex events in the SH, but this is still unclear.

In contrast to observational data, climate model simulations offer the possibility to minimize the influence of internal tropospheric variability by using long-term or ensemble simulations. Yet, models need to be able to simulate both a realistic mean state and variability to be valid tools. However, typical biases in climate models are a stratospheric "cold bias" in the SH, and the resulting excessively strong and persistent stratospheric polar vortex (e.g. Butchart et al., 2011; Charlton-Perez et al., 2013; Lawrence et al., 2022). Models also have biases in the representation of the tropospheric circulation, such as in the position of the mid-latitude jet; this may have consequences for the simulated tropospheric response to stratospheric perturbations, including those induced by ozone depletion and/or climate change (Gerber et al., 2008a; Wilcox et al., 2012; Simpson and Polvani, 2016). However, the implications of these biases on the downward impacts of stratospheric polar vortex extremes in models are not yet fully understood.

In this study, we aim to explore the robustness of the surface impacts of polar vortex anomalies in the SH in reanalysis data and chemistry-climate models (CCMs) in the light of the short observational record and large internal variability. More specifically, we address the following questions: Can CCMs reproduce the surface response of polar vortex anomalies in the SH? Can CCMs help us to identify the robustness of the stratospheric downward impact, given the limited observational record? Which model biases can inhibit a realistic representation of the tropospheric response?

We structure the analysis as follows: after presenting our data and methods in Sect. 2, we show and discuss the results of the stratospheric and surface signals of polar vortex anomalies in Sect. 3.1 and 3.2. We address the variability of the surface signal using bootstrapped surface composites in 3.3. Given the differences in the midlatitude surface signals between reanalysis data and CCMs, we address model biases in Sect. 3.4, revealing opposite biases in terms of the eddy-driven jet latitude and SAM timescales between the models. Finally, we conclude in Sect. 4.

## 2 Data and Methods

### 2.1 Data

For our analysis, we use the reanalysis products MERRA-2 (Gelaro et al., 2017) for the time period 1980-2020, and simulations with the CCMs WACCM version 4 and SOCOL-MPIOM on daily resolution. The variables of interest are zonal mean geopotential height and zonal mean zonal wind, as well as the surface variables sea level pressure, two-meter temperature and total precipitation. All data has been linearly detrended. When averaging over several latitudes, the data are weighted with the cosine of latitude.

MERRA-2, the Modern-Era Retrospective Analysis for Research and Applications version 2, is a global reanalysis dataset produced with the GEOS (Goddard Earth Observing System) atmospheric data analysis system using a 3-dimensional varia-





tional algorithm with a 6-h update cycle. It spans the time frame from 1980 to present and uses a finite-volume dynamical core at a resolution of $0.5° \times 0.625°$ and 72 hybrid-eta levels from the surface to $0.01\,\mathrm{hPa}$ (Gelaro et al., 2017).

SOCOL, the SOlar Climate Ozone Links model, is a coupled CCM that consists of the middle-atmosphere general circulation model MA-ECHAM and the chemistry-transport model MEZON (Stenke et al., 2013) and is coupled to the ocean-sea-ice model MPIOM by the OASIS3 coupler (Muthers et al., 2014). It extends from the earth's surface to $0.01\,\mathrm{hPa}$ (approximately

$80\,\mathrm{km}$) with 39 vertical levels and has a horizontal resolution of spectral truncation T31 ($3.75° \times 3.75°$). The chemistry-transport model includes 41 chemical species, determined by 140 gas-phase reactions, 46 photolysis reactions, and 16 heterogeneous reactions. Chemistry-climate interactions can be disabled by deactivating the coupling between chemistry and dynamics (Muthers et al., 2014). The model captures stratospheric variability reasonably well but shows a cold temperature bias at the pole and overestimates Antarctic total ozone loss during springtime (Stenke et al., 2013).

WACCM version 4, the Whole Atmosphere Community Model is a version of the NCAR Community Earth System Model (CESM1) that resolves the stratosphere and includes interactive chemistry (Marsh et al., 2013). It has 66 vertical levels, a model top at $5.1 \times 10^{-6}\,\mathrm{hPa}$ (approximately $140\,\mathrm{km}$) and a horizontal resolution of $1.9°$ latitude $\times 2.5°$ longitude. The model includes an active ocean and sea ice component with a nominal latitude-longitude resolution of $1°$. The chemistry module is based on the Model for Ozone and Related Chemical Tracers version 3 (Kinnison et al., 2007), and includes a total of 59

chemical species and 217 gas-phase reactions, and 17 heterogeneous reactions on three aerosol types. Like SOCOL, it can be run in a specified chemistry mode with prescribed instead of interactive chemistry (Smith et al., 2014). Stratospheric variability and the development of the ozone hole agree reasonably well with observations but the model shows a cold pole bias (Marsh et al., 2013). The model has been used in several studies analysing stratospheric variability and trends (e.g. Gillett et al., 2019; Haase and Matthes, 2019; Rieder et al., 2019; Oehrlein et al., 2020).

From each model, we use 200-year time-slice simulations that are forced with constant boundary conditions of the year 2000. Seasonally varying greenhouse gas concentrations and ozone depleting substances are fixed to this year. The quasi-biennial oscillation is nudged according to Stenke et al. (2013) in SOCOL, and following Brönnimann et al. (2007) in WACCM. Both models have fully coupled dynamics, radiation and chemistry, and include ozone-circulation feedbacks. As the simulations contain neither climate change nor ozone depletion trends, they are well suited to investigate interannual variability and offer

an unprecedented opportunity to investigate stratosphere-troposphere coupling under near-present day conditions with a larger sample size than in the observations.

## 2.2    Methods

In the Southern Hemisphere, different methods have been used to identify strong and weak polar vortex events in previous studies. We choose a similar detection method as in Thompson et al. (2005) based on the $10\,\mathrm{hPa}$ SAM. The SAM index

is defined according to method 3 in Baldwin and Thompson (2009) as the principal component time series normalized to unit variance from the first empirical orthogonal function of daily, zonal mean geopotential height anomalies south of $20°$S. Latitudinal weighting is applied as the square root of the cosine of latitude. The SAM index is calculated separately for all





pressure levels and by convention, a negative SAM index corresponds to positive geopotential height anomalies over the polar cap and weaker westerly zonal flow and vice versa for a positive SAM index.

As the SAM variance peaks in austral spring, we detect the largest and smallest anomalies in the daily $10\,\mathrm{hPa}$ SAM index between August and November each year. From these values, we define the highest and lowest $25\,\%$ as the strong and weak polar vortex events. Therefore, we obtain 10 strong/weak polar vortex events in the reanalysis data and 50 strong/weak events in the CCMs. For strong and weak polar vortex composites, we define onset dates as the time when the SAM value crosses +2, -2 standard deviations respectively, prior to the peak magnitude of the event (Thompson et al., 2005). The onset and peak

timing and magnitudes of the SAM index are documented for MERRA-2 in Table A1 in the Appendix.

The annular mode timescale is an integrated measure of annular mode variability and serves as an estimate of the persistence of annular mode anomalies (Gerber et al., 2010). We compute the SAM timescale as a function of season and height to quantify the persistence of SAM anomalies in the stratosphere and troposphere. SAM timescales are measured by the lag time (in days) that the SAM autocorrelation function takes to drop to $1/e$. For the calculation, we use the methods described in Gerber et al.

(2008b) and Simpson et al. (2011) for the SAM index by performing the following steps on all pressure levels:

1. The autocorrelation function (ACF) of the SAM is calculated for every day of the year $d$ and lag $l$ using the function

$$ACF(d,l) = \frac{\sum\limits_{y=1}^{N_y-1} SAM(d,y)SAM(d+l,y)}{\sqrt{\sum\limits_{y=1}^{N_y-1} SAM(d,y)^2 \sum\limits_{y=1}^{N_y-1} SAM(d+l,y)^2}} \tag{1}$$

where $y$ is the year and $N_y$ the number of years.

2. The ACFs are then smoothed with a Gaussian filter with standard deviation $\sigma = 18$.

3. The e-folding timescale $\tau$ is estimated by applying a least square fit of the exponential function $e^{-l/\tau_N}$ to the ACF up to lag $l = 50$ (Gerber et al., 2008a).

We define a daily jet latitude index of the tropospheric eddy-driven jet as the location of the maximum $850\,\mathrm{hPa}$ zonal mean zonal wind between 35°S and 70°S (Byrne et al., 2019), interpolated to a latitudinal grid of 0.1°. Note that we use the terms midlatitude jet and eddy-driven jet interchangeably.

Daily anomalies are calculated by subtracting the climatology of each day of the year, which is computed by averaging over all available years for each calendar day. The climatology is therefore calculated for the period 1980-2020 in the reanalysis data, and over the 200 model years in the CCM simulations.

We perform a 1-sample bootstrapping test to estimate the significance of the time-height and surface composites. The composites based on the detected polar vortex events are compared to a distribution of 1000 random composites. These are created

by sampling random years for the central dates in the time-height composites, and random October to January periods for the surface composites, respectively. The actual composite is significantly different from 0 when it differs more than 2 standard deviations from the mean of the random distribution, which corresponds to a significance level of $95.5\,\%$.



To estimate the uncertainty of the surface signal and to what extent it is affected by sampling variability, we use the boot-strapping method of Deser et al. (2017); Oehrlein et al. (2021). The observed composite consists of 10 events with tropospheric
states that are unrelated to the stratospheric signal. We randomly resample the 10 observed events with replacement to form 500 synthetic composites. In the synthetic composites, we allow an individual event to be repeated a maximum of three times. We thereby estimate how much the surface signal varies between the synthetic composites and how it relates to the strength of the polar vortex anomaly. Similarly, we generate 500 synthetic composites of the CCMs, randomly sampling 10 out of the 50 weak polar vortex years to obtain comparable composites as in the observations and also assess their relationship to the
strength of the stratospheric perturbation.

We focus on the regional surface responses in Australia and Antarctica, as these are shown to be affected by polar vortex anomalies. We assess the variability across composites in these regions and calculate the area-weighted average for:

1. Surface temperature in Antarctica: 65°-90° S, excluding the Antarctic Peninsula region: 30°W-100°W

2. Surface temperature in Australia: 20°-40°S, 113°-154°E

## 3 Results and Discussion

### 3.1 Stratospheric Signal

To explore the range of the SH polar vortex variability in spring, we analyze the $10\,\mathrm{hPa}$ SAM. The SAM is directly related to zonal mean wind and is thus a valid metric of the polar vortex strength. The $10\,\mathrm{hPa}$ SAM distributions of the $25\,\%$ largest positive and negative daily polar vortex anomalies from August to November are shown in Fig. 1. The most extreme strong
polar vortex anomaly in the observations occurred in 2020, when the SAM index exceeded 4 standard deviations. The weakest polar vortex events in the observations occur in 2002 and 2019, with SAM values below -9 standard deviations. Neither model reproduces the two most negative SAM events in reanalysis, although the SAM extremes in the models get reasonably close (-7 in WACCM and -8 in SOCOL). The inability of models in capturing extremes beyond 9 standard deviations may be due to the strong polar vortex bias in both CCMs. In WACCM, part of the reason may also be the weaker tropospheric wave forcing,
as revealed by the smaller $100\,\mathrm{hPa}$ eddy heat flux anomalies near the onset date in this CCM (Fig. A3). That said, both CCMs do reproduce the broad features of stratospheric SAM variability reasonably well, namely the asymmetry in SAM extremes, with larger negative anomalies, and the bulk of the weak events overlap observations.

We create composites of the SAM indices to compare the time and height evolution of the weak and strong polar vortex events between the reanalysis and CCMs, shown in Fig. 2. The anomalies peak in the mid- to upper stratosphere following the onset
date (by construction) and propagate down to the lower stratosphere, where they persist for up to 90 days (Fig. 2 a,b), consistent with similar previous observational analyses (Thompson et al., 2005; Byrne and Shepherd, 2018). Stratosphere-troposphere coupling is apparent in all datasets as the tropospheric signal after the onset of the peak stratospheric vortex anomaly tends to be of the same sign as the lower stratospheric anomaly. The time period of statistically significant downward impact is intermittent and does not exactly match between the datasets. While for weak events, for example, the tropospheric signal





peaks at days 30-60 in MERRA-2, the stratosphere-troposphere coupling is stronger and longer lived in SOCOL, resulting in a tropospheric signal that is more significant and persistent, while the opposite is seen in WACCM, which exhibits weaker and shorter-lived tropospheric anomalies.

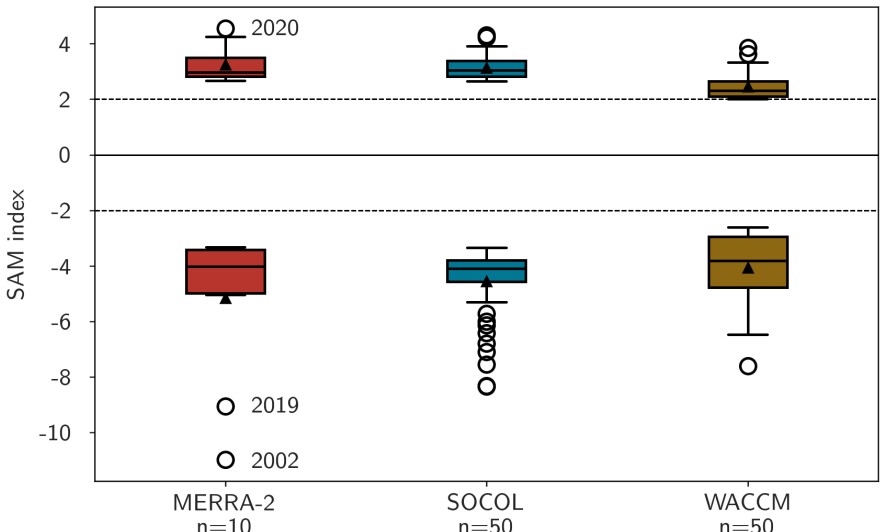

**Figure 1.** The $10\,\text{hPa}$ SAM index values of the $25\,\%$ lowest and highest springtime SAM indices in MERRA-2 (10 events per weak/strong category), and in CCMs WACCM and SOCOL (50 events per weak/strong category). The most extreme events in the observations are annotated with the year of occurrence.

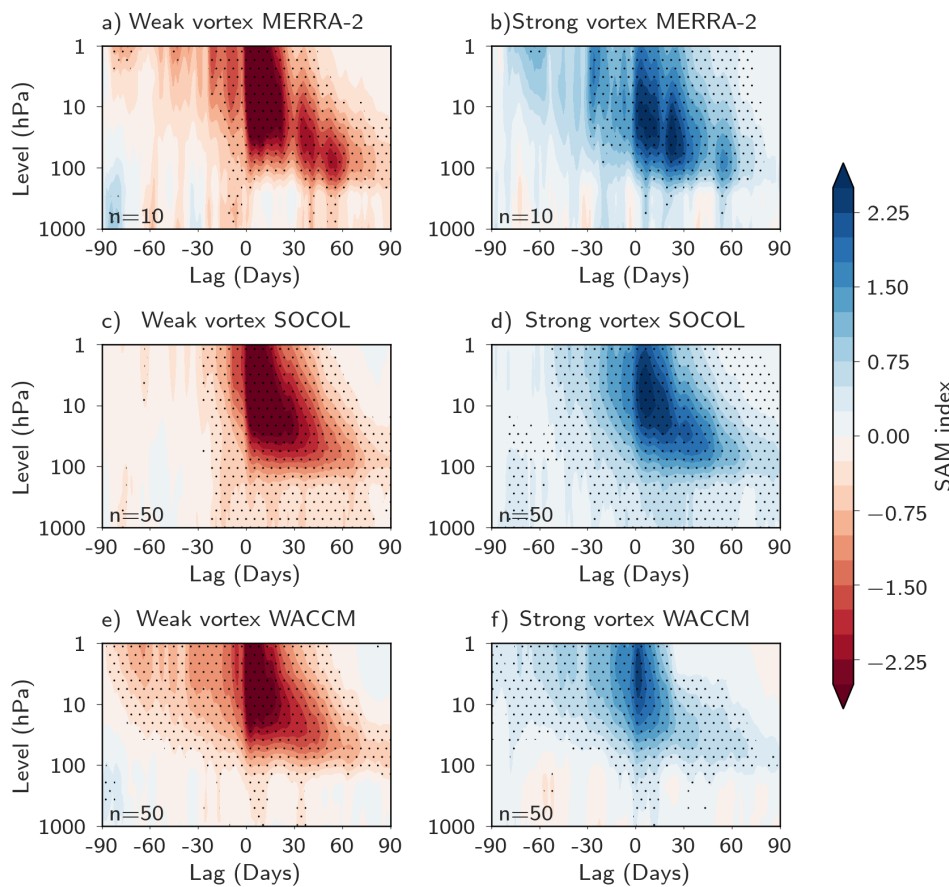

**Figure 2.** Time-height development of the SAM index following weak and strong polar vortex events for the reanalysis MERRA-2 (a,b), and the CCMs SOCOL (c,d) and WACCM (e,f). The central date (lag 0) refers to the first day when the $10\,\mathrm{hPa}$ SAM values have fallen below -2, above +2, respectively. The indices are non-dimensional and stippling refers to significance at the $4.5\,\%$ level assessed with a bootstrapping test.





### 3.2 Tropospheric SAM and Surface Climate

In the time-height development of weak and strong polar vortex events, we have identified that the persistent weak and strong
anomalies in the lower stratosphere are associated with a tropospheric SAM of the same sign in the reanalysis and CCMs. We
now further investigate the impact of the polar vortex extremes on the troposphere and surface climate in the austral spring-
summer season from October to January as in Lim et al. (2018, 2019). We use the term "regime" to refer to the tropospheric
pattern emerging after the weak and strong polar vortex events.

To examine how the tropospheric SAM differs between weak and strong polar vortex events, we compare the distribution
of the average October to January SAM index at $500\,\mathrm{hPa}$ between models and reanalysis (Fig. 3). The tropospheric SAM is
on average negative following weak polar vortex events, in contrast to a positive SAM index following strong polar vortex
events. In all datasets, the distributions of weak and strong polar vortex regimes are significantly different from each other
based on a two-sided Student's t-test on a $5\,\%$ level. However, the magnitude of the SAM response for both weak and strong
polar vortex regimes differs among datasets. In the reanalysis, there is an asymmetry in the response with a more strongly
negative average SAM index during weak polar vortex regimes as compared to the magnitude of the SAM anomalies for strong
polar vortex regimes, which is consistent with the asymmetry in stratospheric anomalies and therefore downward coupling
(Fig. 1). In SOCOL, the average SAM index for weak and strong regimes is more symmetric and of higher amplitude than in
the reanalysis. The SAM response in WACCM is much smaller for both weak and strong polar vortex years. The larger shift
in the tropospheric SAM distributions in SOCOL compared to all other datasets is consistent with the stronger stratosphere-
troposphere coupling in this model seen in the time-height development in Fig. 2. Conversely, the smaller shift in WACCM is
consistent with the weaker stratosphere-troposphere coupling in this model.

Since the tropospheric SAM is known to modulate the surface climate in the SH midlatitude and polar regions (e.g. Hendon
et al., 2007), we examine the surface patterns in October-January following stratospheric anomalies in the reanalysis and the
CCMs. We primarily focus on weak polar vortex events, for which the observed tropospheric SAM response is larger than
for strong polar vortex events, and which are associated with surface extremes in Australia in the following spring to summer
(Lim et al., 2019). Anomalies associated with weak polar vortex regimes in sea level pressure (SLP), surface temperature, and
precipitation are shown in Fig. 4 (strong polar vortex regimes are shown in Fig. A4 in the Appendix). The SLP composites
show a large-scale pattern with positive pressure anomalies over Antarctica and negative pressure anomalies in midlatitude
regions, consistent with the negative phase of the SAM displayed in Fig. 3. However, the magnitude and spatial extent of the
SLP signal differs among the datasets, with a much weaker signal in WACCM than in SOCOL, consistent with the differences
among these models in their tropospheric SAM response (Fig. 3). Despite these differences, it is remarkable that both models
and observations are consistent in exhibiting warm anomalies over Antarctica of up to $1\,\mathrm{K}$.

In the midlatitudes, the surface signature of weak polar vortex events in temperature, SLP, and precipitation in MERRA-2 is
remarkably similar to previous studies (Lim et al., 2018, 2019), despite using a simpler methodology in this paper (note that in
our study, we calculate the SAM index at every level independently and thus do not take the vertical covariance into account).
However, MERRA-2 and CCM simulations differ both in the magnitude and sign of the anomalies. For example, CCMs do not

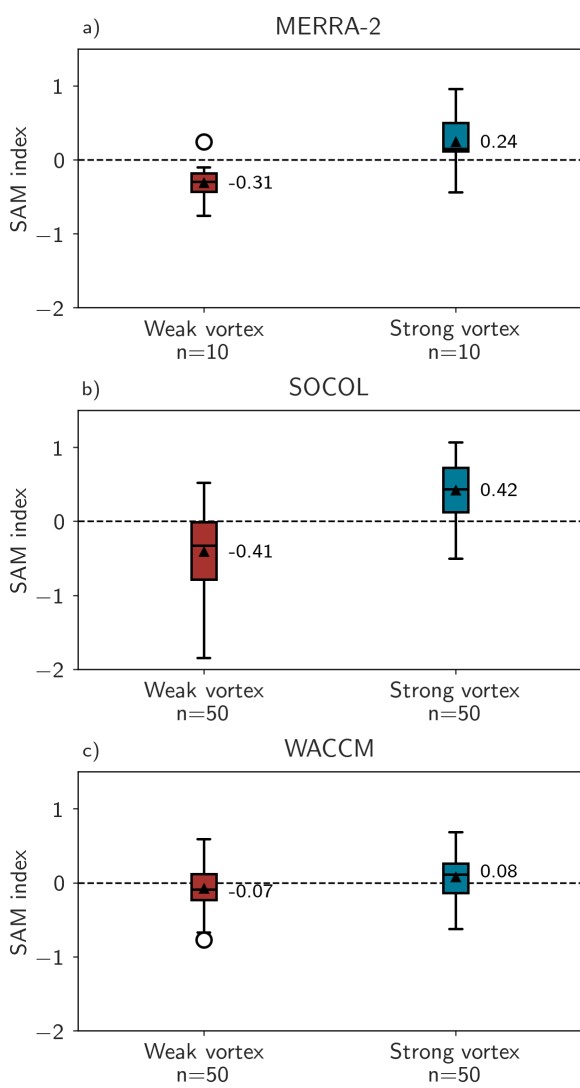

**Figure 3.** Distribution of the mean $500\,\text{hPa}$ SAM index averaged over the October-January period following weak and strong polar vortex anomalies for MERRA-2 (a), SOCOL (b), and WACCM (c). The box extends from the lower to upper quartile of the data, the whiskers extend from the lower quartile $-1.5$ IQR to the upper quartile $+1.5$ IQR. Data points outside of the whiskers are shown as circles. The horizontal line marks the median value and the triangle indicates the mean of the distribution, which is annotated next to the box.





show the warm and dry anomalies over southern and eastern Australia that are visible in MERRA-2. On the contrary, SOCOL shows cold temperature anomalies in southern Australia, while no significant signal is visible in WACCM over Australia and generally in the midlatitudes.

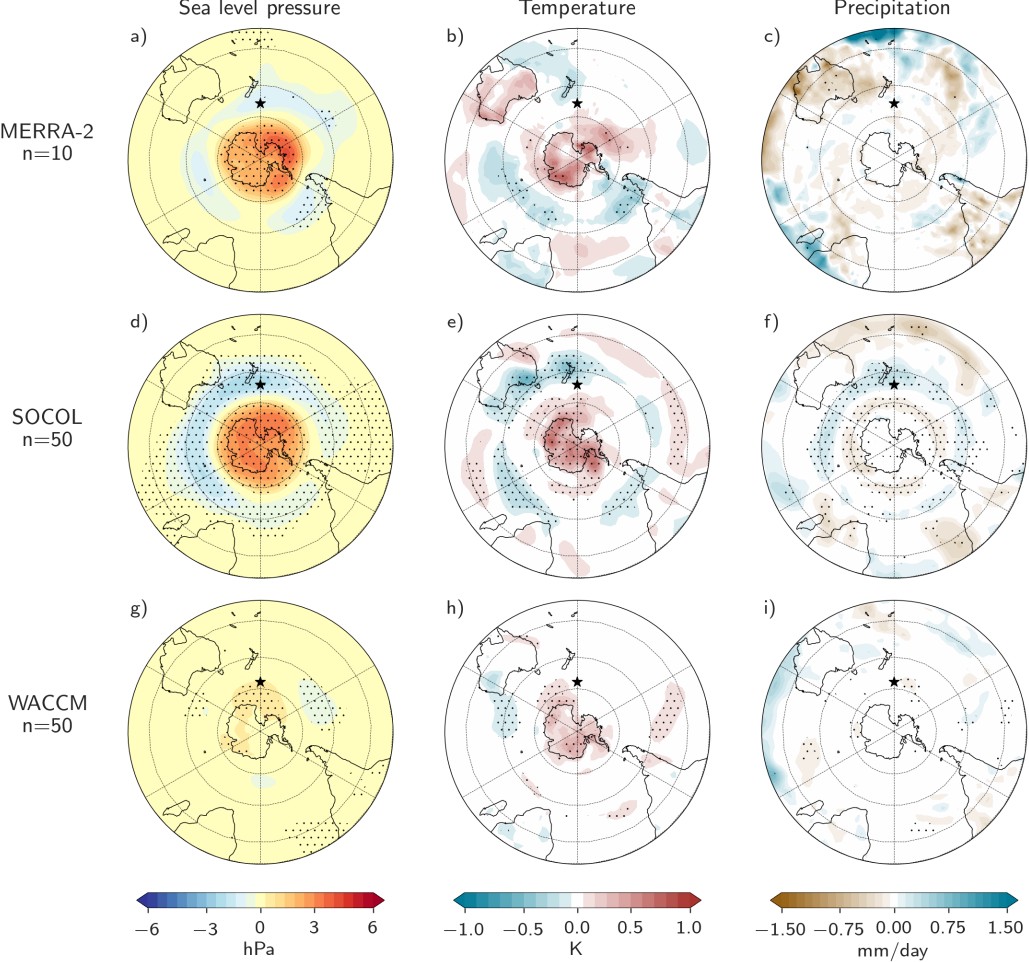

**Figure 4.** Surface climate composite for weak polar vortex regimes of October-January SLP anomalies (a,d,g), 2-meter temperature anomalies (b,e,h) and precipitation anomalies (c,f,i). The reanalysis data MERRA-2 (first row) includes 10 weak vortex regimes, and the CCMs SOCOL (second row) and WACCM (third row) each include 50 weak vortex regimes. Stippling refers to significance on a $4.5\%$ level assessed with a bootstrapping test.





### 3.3 Uncertainty in the Surface Climate Response

In the previous Sections, we have shown that some features of the modeled tropospheric and surface signals following weak polar vortex events do not correspond to the observations, especially in the midlatitudes. However, these signals may also be influenced by internal variability unrelated to the stratospheric forcing. The question arises of how robust the observed tropospheric signal is, given the small number of observed events (n=10); hence, differences between the reanalysis data and CCMs could be due to a sampling issue. Fig. 5 shows selected examples of surface temperature composites of random subsamples with 10 out of the 50 weak polar vortex regimes, to be consistent with the sample size in reanalysis. In these subsamples, it becomes evident that the more zonally symmetric anomalies in the CCM composites depicted in Fig. 4 arise from averaging over a larger sample size in the models. In the examples shown in Fig. 5, the warming anomaly over Antarctica is robust across reanalysis and models. Conversely, the temperature anomaly over Australia varies among subsamples; in some of these subsamples, both models reproduce the observed warm anomaly (panels d, e), whereas in others the models show cooling instead of warming (b, c).

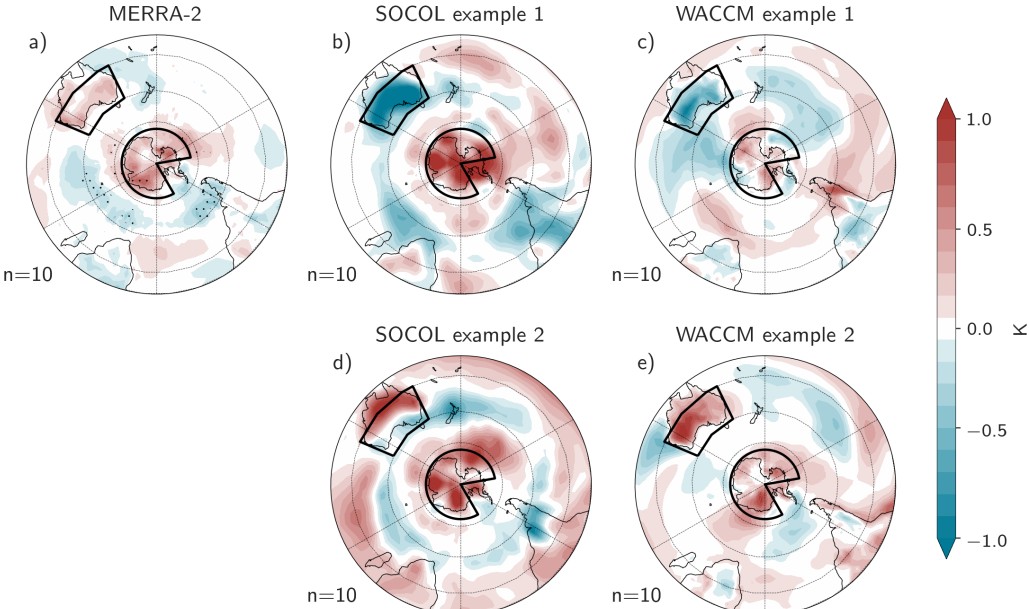

**Figure 5.** Examples of subsampled (n=10) temperature anomaly composites of weak polar vortex regimes for October to January for SOCOL (b,d) and WACCM (c,e) with the MERRA-2 (n=10) composite (a) for comparison.

We further examine the robustness of the observed and simulated surface response to sampling variability in targeted regions (Australia and Antarctica, as marked in Fig. 5) and construct synthetic composites by randomly sampling 10 of the observed composites with replacement, and subsampling 10 of the 50 simulated weak polar vortex composites, following the same procedure of Oehrlein et al. (2021). Fig. 6 shows scatterplots of Australian and Antarctic temperatures and the stratospheric





polar vortex strength (in terms of SAM anomalies at $10\,\mathrm{hPa}$) in the 500 bootstrapped composites (n=10) in reanalysis and CCMs. To the right of the scatterplot panels, the PDFs of the temperature distributions are shown for all 500 bootstrapped composites and all datasets.

We begin with the Antarctic temperature anomaly in Fig. 6 a-d. In the reanalysis data, the sign of Antarctic temperature
240    anomaly in weak polar vortex regimes is robust with $99\,\%$ of resampled composites showing warming, but there is a large spread in magnitude (Fig. 6 b). The mean of the resampled composites is $0.3\,\mathrm{K}$ with a standard deviation of $0.12\,\mathrm{K}$. In the models, most subsampled composites also show Antarctic warming as a response to polar vortex weakening ($95\,\%$ in SOCOL and $85\,\%$ in WACCM), highlighting the robustness of the warming signal over Antarctica after weak vortex events. Yet, the magnitude of this positive anomaly is subject to uncertainty, given the large spread in the magnitude of the subsamples in the
245    reanalysis data as well as in the CCMs.

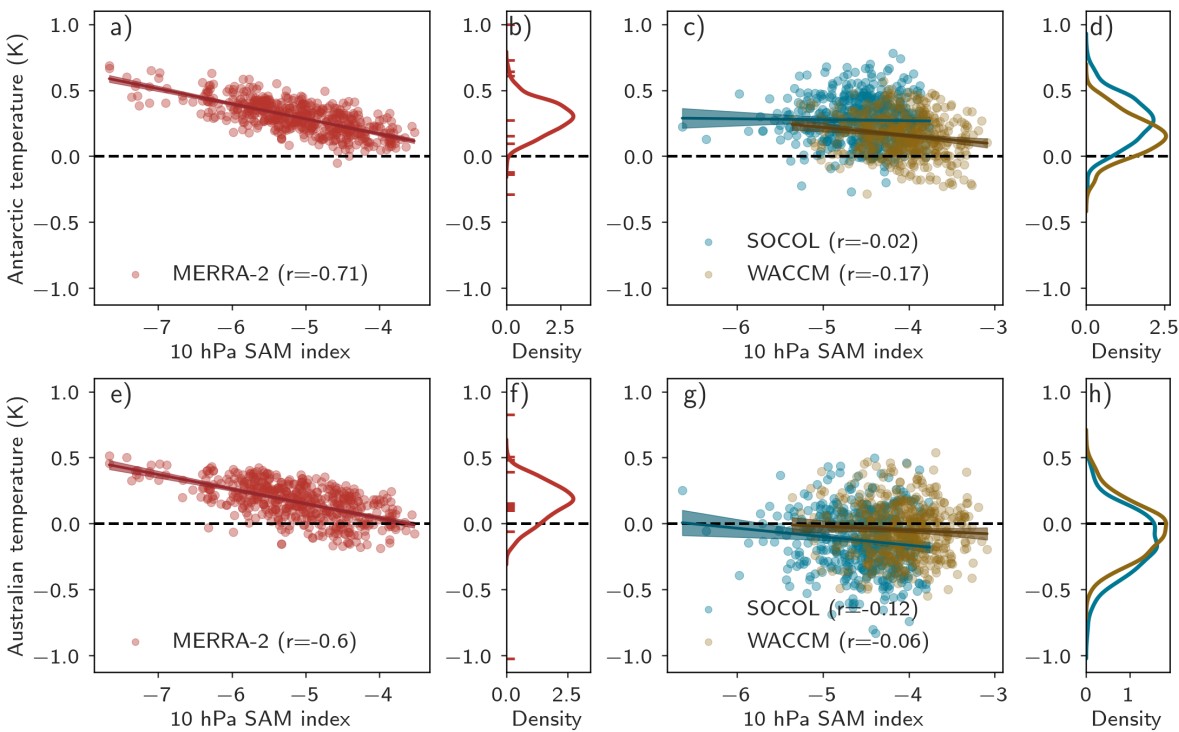

**Figure 6.** Scatterplots of 500 synthetic bootstrapped composites (n=10) for Antarctic (a,c) and Australian (e,g) 2-meter temperature anomalies in weak polar vortex regimes vs. the composite stratospheric $10\,\mathrm{hPa}$ SAM peak anomaly. The *r*-value refers to the Pearson correlation coefficient between the two quantities, the line refers to the fitted linear regression with the $95\,\%$ confidence interval of the slope in shading. The kernel density estimation (KDE) of the temperature composites is shown on the right of the scatterplots (b,d,f,h), and the mean $2\,\mathrm{m}$ temperature values for the 10 weak polar vortex regimes are marked with red dashes for the reanalysis datasets.





In Australia, where $86\%$ of bootstrapped composites in MERRA-2 show warming as a response to polar vortex weakenings, there is a large variety in the temperature response in the CCMs. The average signal in WACCM is $0.05\,\mathrm{K}$ with a standard deviation of $0.2\,\mathrm{K}$. In SOCOL, the average of the synthetic composites is negative with a mean of $-0.13\,\mathrm{K}$ and a similar spread as in WACCM with a standard deviation of $0.22\,\mathrm{K}$. While the sign of the temperature response over Australia is largely robust in reanalysis, the two models, on average, do not reproduce the observed warming following SH vortex weakenings.

The negative correlation between the Antarctic and Australian temperature anomalies and the stratospheric SAM in MERRA-2 shows that some of the variability in the surface temperature response is explained by the strength of the stratospheric perturbation. This strong correlation between the surface signal and the magnitude of the stratospheric anomaly raises the question of how much the surface and bootstrapped composites are influenced by the two most extreme events, namely 2002 and 2019. When excluding these events, $95\%$ of the bootstrapped composites for the Antarctic surface signal still show a positive anomaly but the negative correlation with the stratospheric peak amplitude SAM weakens (r=$-0.33$). This confirms the robustness of the sign of the Antarctic temperature response. Moreover, the uncertainty in the magnitude is related to the strength of the stratospheric anomalies, confirming a downward influence in the observations. Conversely, the magnitude of the stratospheric SAM extreme is only very weakly correlated with the tropospheric signal in the CCMs, which suggests that internal (tropospheric) variability unrelated to polar vortex conditions accounts for most of the spread in the magnitude of the modeled signal. Additional tropospheric variability might be contributed by tropical teleconnections arising from ENSO.

In summary, the bootstrapped composites reveal a large uncertainty in the magnitude of the surface temperature response over Antarctica and Australia in both reanalysis and CCMs. However, while the Antarctic warming is evident in both observations and CCMs, the sign of the Australian temperature signal is only robust in the reanalysis, while there is no robust Australian temperature signal in the models. This suggests that the large differences in the surface patterns between observations and models, as shown in Fig. 4, are unlikely to solely result from the short observational record and the limited number of observed SH vortex weakenings. Rather, differences between the PDFs of the bootstrapped temperature composites over Australia in Fig. 6 indicate systematic differences between the reanalysis data and the CCMs. One possible reason for these disagreements between models and observations are model biases, as assessed next.

## 3.4 Role of Model Biases in the Simulated Surface Climate Response

Given the differing surface impacts in both magnitude and regional extent between reanalysis data and CCMs, we first examine metrics characterizing the background state that are relevant for stratosphere-troposphere coupling.

We start with the climatology of the polar vortex and use $10\,\mathrm{hPa}$ zonal mean zonal wind at $60°\mathrm{S}$ as a representation of the vortex strength. The mean annual cycle of the polar vortex is shown in Fig. 7 a) for MERRA-2 and the two CCMs. In both CCMs, the polar vortex shows a bias towards stronger zonal mean zonal winds and a later transition of westerly to easterly winds in spring. The strong polar vortex bias is particularly pronounced from June to January in WACCM. In this model, we even find years with year-round westerlies with no transition to easterly winds. The stronger westerly wind velocities in the models are likely a reason for the smaller amplitude of weak vortex events (Fig. 1), as less planetary waves can propagate upward (Charney and Drazin, 1961).



Consistent with a strong vortex bias, the CCMs show very low stratospheric ozone variability (Fig. A2). Ozone feedbacks have also been suggested to be relevant for surface impacts (Hendon et al., 2020). In our models, the inclusion of interactive ozone has a significant impact on the evolution of the stratospheric SAM (not shown), but does not lead to significant differences in the troposphere (Fig. A1), suggesting that ozone feedbacks are not important for the surface response following stratospheric vortex anomalies in these two CCMs (see Appendix Sect. A1). The smaller ozone variability in the models is thus unlikely to

explain the inability of the models in reproducing some of the observed surface signals reported above.

The SAM in the troposphere is characterized by meridional vacillations of the midlatitude jet location (Thompson and Wallace, 2000), and is influenced by stratospheric variability (Fig. 2, and e.g. Thompson et al., 2005). Moreover, the latitude of the tropospheric eddy-driven jet can also affect the strength of stratosphere-troposphere coupling, as shown in idealized model experiments (Garfinkel et al., 2013). By inspecting the climatological latitude of the eddy-driven jet, we seek to find reasons for

the disagreement between modeled and observed surface patterns following weak polar vortex events in Fig. 4; this is shown in Fig. 7 b. It is readily apparent that there are opposite biases in the two models: in WACCM, the midlatitude jet is biased south in contrast to a northward bias in SOCOL. While SOCOL deviates more from the reanalysis in the winter months than WACCM, it agrees better in the spring-summer season, which is the relevant time period for stratosphere-troposphere coupling in the SH. The jet in WACCM barely shows the equatorward migration in the summer season, which is better represented in

SOCOL.

Model biases in the jet position are consistent with the CCMs' tropospheric SAM surface patterns (Fig. A5) differing from those of the reanalysis. Nevertheless, despite the CCMs' biases, the polar vortex perturbations project on the tropospheric SAM, which is in line with the SAM response in very simple models (e.g. Domeisen et al., 2013).

Another indicator for tropospheric biases affecting the downward response from the stratosphere is the persistence of the

SAM, which is represented by the SAM timescales. This metric provides useful insights into the model skill in representing low-frequency variability in the atmospheric circulation (Gerber et al., 2008b). An overestimated annular mode timescale implies that the modeled circulation may be overly sensitive to external forcings. Conversely, a short annular mode timescale in the troposphere is related to a small downward influence of the stratosphere (Gerber et al., 2008a; Chan and Plumb, 2009; Son et al., 2010). We show the SAM timescales as a function of season and pressure level in Fig. 8. Generally, anomalies in

the SAM decay more slowly (and thus the timescale is longer) in the stratosphere than in the troposphere. While the models capture the general seasonal cycle of the SAM timescale, the stratospheric and tropospheric maxima are delayed compared to the reanalysis. The delayed seasonal cycle likely results from the strong vortex bias. Additionally, both models show a late spring polar vortex breakup compared to the observations, as seen in Fig. 7 a, which might delay the seasonal cycle in the troposphere. Most remarkably, the SAM timescales in CCMs differ in opposite ways with respect to the observations in the

troposphere, where SAM timescales are strongly overestimated in SOCOL, a typical bias of climate models in the SH (Gerber et al., 2010). In contrast, tropospheric SAM timescales in WACCM are shorter than in the reanalysis, particularly in spring to summer.

The opposite biases of tropospheric SAM timescales in the CCMs are consistent with their different eddy-driven jet locations (Fig. 7 b). Climatological jet locations and SAM timescales are shown to be highly correlated, with lower SAM timescales



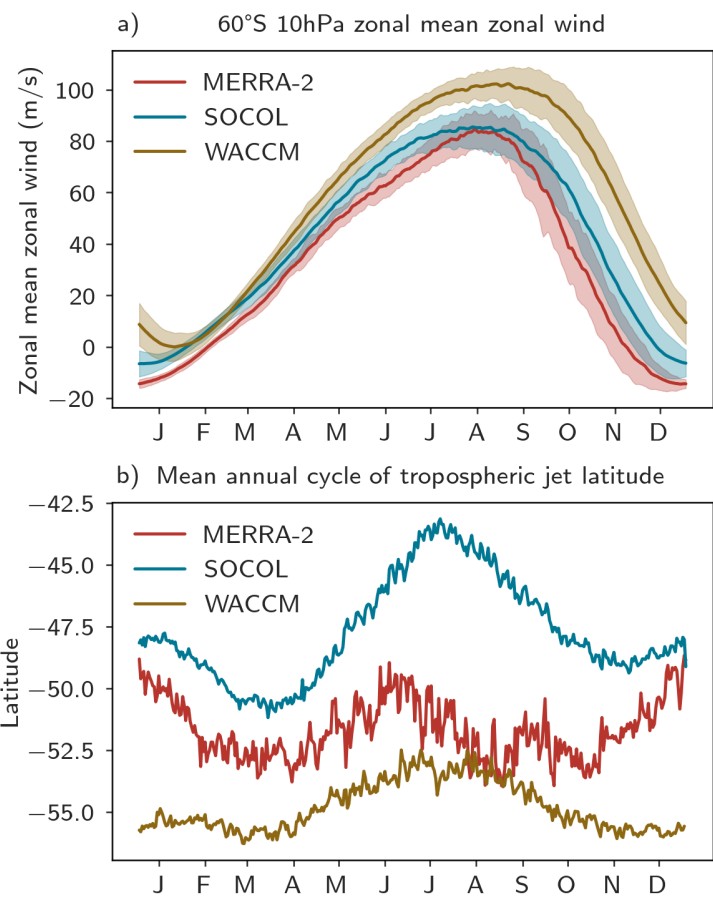

**Figure 7.** Mean annual cycle of the 10 hPa zonal mean zonal wind at 60°S with standard deviation (shading) (a), and the mean annual cycle of daily jet latitude indices defined as the location of the maximum 850 hPa zonal mean zonal wind between 35°S and 70°S for MERRA-2 and the CCMs WACCM and SOCOL (b).

for jet locations at higher latitudes (Son et al., 2010; Kidston and Gerber, 2010). The differing SAM timescales are related to eddy-mean flow feedbacks that are sensitive to the latitude of the eddy-driven jet (Son et al., 2007; Gerber and Vallis, 2007; Simpson et al., 2010). For example, eddy activity is confined to a relatively small latitudinal band of high baroclinicity at the edge of the Hadley cell for a more equatorward jet, which can make zonal mean flow anomalies more persistent.

     Taken together, we have identified model biases in the tropospheric circulation, which are likely the reason for the dis-
agreement between models and observations, namely the overestimation in the tropospheric response in SOCOL, and the underestimation in WACCM in comparison with reanalysis data.



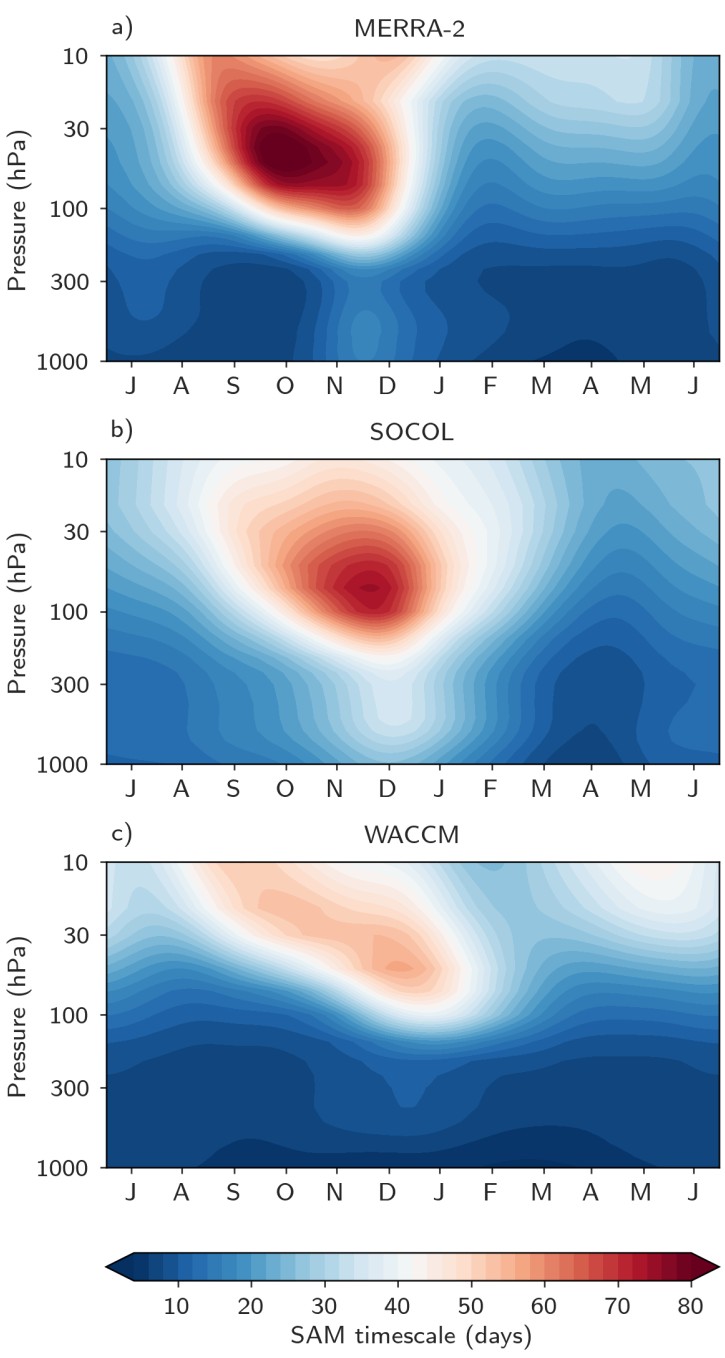

**Figure 8.** The SAM timescale $\tau$ (in days) as a function of season and height in the reanalysis MERRA-2 (a) and the models SOCOL (b) and WACCM (c).





## 4 Conclusions

In this study, we assess the role of interannual austral stratospheric vortex variability in forcing spring- and summertime surface climate. Based on the analysis of observational data and targeted CCM simulations, we have examined the downward impact of
polar vortex anomalies on interannual timescales in the spring-summer season (October-January), confirming previous findings (e.g. Thompson et al., 2005; Lim et al., 2018, 2019; Kwon et al., 2020). The main results are as follows:

- The downward impact of the polar vortex can be seen in the subsequent shift of the tropospheric SAM to its negative/positive phase in weak/strong polar vortex regimes. Further, our observational analysis confirms the surface response of weak polar vortex regimes reported previously with warming and dry conditions over Antarctica and Australia. How-
ever, our results also show that while models robustly capture the warming signal over Antarctica, they struggle to reproduce the observed surface signal in the midlatitudes, especially over Australia.

- An "observational large ensemble" analysis based on a bootstrapping method reveals that the observed warming signal over Antarctica and Australia is robust, although the magnitude of the signal is uncertain. In the model experiments, we find that the Australian temperature signal is even more uncertain than the warming over Antarctica, with equal
likelihoods of warming and cooling. Despite the short observational record and thus limited number of observed vortex weakenings in the SH, the reanalysis data reveal a surface signal that is more robust in its sign and more correlated to the stratospheric forcing than in the long-term modelling experiments. Thus, we exclude internal variability as a single reason for differences in surface signals between models and observations.

- Biases in the polar vortex strength, eddy-driven jet location and SAM timescales limit the models' ability to capture
observed signals in midlatitudes. The bias in the surface impact of stratospheric circulation anomalies differs between models, with WACCM possibly underestimating and SOCOL overestimating the downward stratospheric impacts. It is suggested that this is due to biases in the latitude of the tropospheric jet and the SAM timescale, with WACCM having a poleward bias in the jet and a too short timescale, whereas the jet is biased equatorward and the SAM timescale is too long in SOCOL.

While understanding of stratosphere-troposphere coupling and associated surface impacts have advanced in recent times, further research is necessary to gain a better understanding of the relevant processes and their representation in numerical models. Improving the representation of SH large-scale dynamics in the stratosphere and troposphere in models, as well as dynamical and ozone variability, is important to further investigate surface climate impacts associated with stratospheric forcings. Considering the ongoing changes in the stratosphere, with ozone recovery and increasing greenhouse gas concentrations,
further work is necessary to better understand stratosphere-troposphere coupling and how it may change in the future, both on long-term and interannual timescales.





## Appendix A

### A1   Ozone Feedbacks

The SH tropospheric circulation is also known to be sensitive to stratospheric ozone variations. Long-term ozone depletion has driven widespread surface climate changes (e.g. Thompson and Solomon, 2002; Thompson et al., 2011; Previdi and Polvani, 2014). Aside from long-term changes, a downward influence has also been suggested from the interannual variability of stratospheric ozone in spring to summertime surface climate (Son et al., 2013; Bandoro et al., 2014; Gillett et al., 2019; Damiani et al., 2020). However, dynamical and ozone variability are strongly linked and separating ozone feedbacks from dynamical variability is difficult. Including ozone feedbacks in weather and climate models may result in more accurate results, as it for example has been shown for the 2002 SSW in the SH (Hendon et al., 2020). However, interactive ozone is also computationally expensive.

To isolate the influence of ozone-circulation feedbacks, we compare simulations with fully interactive ozone to those with specified ozone chemistry. In the fully interactive ozone simulations (INT-O3), the free running models interactively calculate ozone concentrations, which allows direct feedbacks with radiation and dynamics. The runs with specified ozone chemistry (CLIM-O3) still interactively calculate ozone in the background, but ozone is decoupled from the radiation scheme and replaced with monthly mean, zonal mean ozone climatologies derived from the 200-year long INT-O3 runs. For both models, the INT-O3 and CLIM-O3 simulations have each 200 model years.

In Fig. A1, we show the average tropospheric SAM in weak and strong polar vortex regimes from the simulations with interactive (as in Fig. 3), as well as climatological ozone. On average, model simulations with climatological ozone (which by definition do not include radiative/dynamical feedbacks from ozone) also show a negative tropospheric SAM during weak polar vortex regimes, and a positive SAM during strong polar vortex regimes in October-January, similar to simulations including fully interactive ozone chemistry. We find small but significant differences in the magnitude and persistence of SAM anomalies between simulations including and excluding ozone feedbacks in the stratosphere (not shown). Conversely, ozone feedbacks have little effect on the tropospheric SAM signal and surface climate.

Taken together, these results suggests a dominant role of dynamical variability for stratospheric polar vortex extremes and their downward influence on tropospheric and surface climate, while ozone feedbacks only play a minor role in the downward coupling. However, the CCMs underestimate ozone variability, as shown in Fig. A2, possibly resulting in an underestimation of ozone feedbacks. Hendon et al. (2020) shows the importance of stratospheric ozone for accurately simulating anomalies in the stratosphere and at the surface for the 2002 SSW. However, 2002 is the most extreme event in the observations and the absence of such high amplitude perturbations in the CCMs may explain the missing contribution of ozone feedbacks to surface climate in the CCMs.





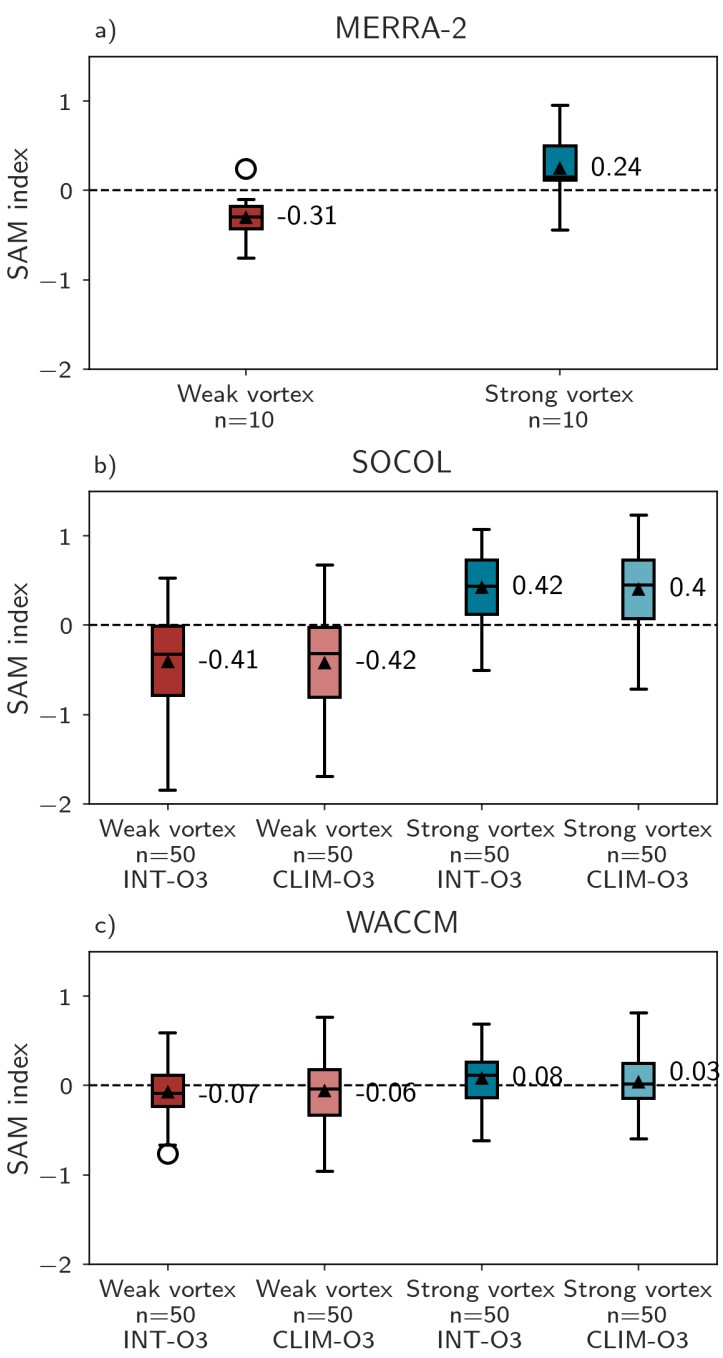

**Figure A1.** Distribution of the mean $500\,\mathrm{hPa}$ SAM index averaged over the October-January time period following weak and strong polar vortex anomalies for MERRA-2 (a), SOCOL INT-O3 and CLIM-O3 (b) and WACCM INT-O3 and CLIM-O3 (c). The box extends from the lower to upper quartile values of the data, the whiskers extend from the lower quartile $-1.5$ IQR to the upper quartile $+1.5$ IQR. Data points outside the whiskers are shown as a circle. The horizontal line marks the median value and the triangle the mean of the distribution, which is annotated next to the box.





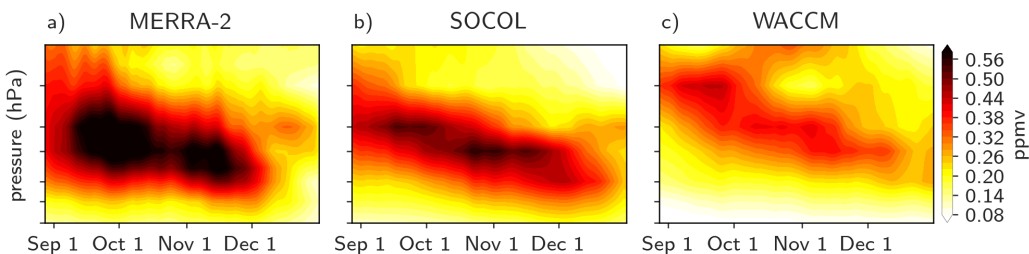

**Figure A2.** Ozone standard deviation in austral spring in MERRA-2 (a), SOCOL (b) and WACCM (c).





## A2    Supplementary Information

In this Section, we show the onset and peak dates as well as the peak amplitude of polar vortex events in MERRA-2 in Table A1. Additional Figures include the eddy heat flux composite for MERRA-2 and the CCMs in Fig. A3, the surface climate composites of strong polar vortex regimes for surface pressure, temperature, and precipitation anomalies in Fig. A4 and the regression of 2-meter temperature on the $1000\,\mathrm{hPa}$ SAM in Fig. A5.

**Table A1.** Details on timing and magnitude of the detected weak and strong polar vortex events in MERRA-2 used in this study. Peak amplitude is in standard deviation and refers to the SAM index at $10\,\mathrm{hPa}$.

| | Weak polar vortex | | | | Strong polar vortex | | |
|---|---|---|---|---|---|---|---|
| Year | Onset date | Peak date | Peak amplitude | Year | Onset date | Peak date | Peak amplitude |
| 1982 | Oct 20 | Oct 22 | -3.7 | 1987 | Oct 13 | Nov 16 | 4.6 |
| 1988 | Oct 24 | Oct 31 | -4.8 | 1996 | Nov 04 | Nov 06 | 2.9 |
| 1992 | Oct 18 | Oct 20 | -3.5 | 1997 | Oct 21 | Oct 30 | 3.1 |
| 2000 | Oct 17 | Oct 28 | -3.5 | 1998 | Nov 20 | Nov 27 | 3.0 |
| 2002 | Sep 21 | Sep 27 | -10 | 1999 | Nov 10 | Nov 13 | 3.0 |
| 2004 | Oct 9 | Oct 20 | -3.4 | 2006 | Nov 4 | Nov 6 | 2.7 |
| 2007 | Sep 19 | Sep 20 | -3.4 | 2008 | Nov 16 | Nov 21 | 2.7 |
| 2012 | Oct 7 | Oct 16 | -5.0 | 2010 | Nov 8 | Nov 18 | 2.8 |
| 2013 | Oct 13 | Oct 21 | -4.4 | 2015 | Oct 8 | Nov 02 | 3.6 |
| 2019 | Aug 30 | Sep 20 | -9.1 | 2020 | Oct 5 | Nov 27 | 4.2 |





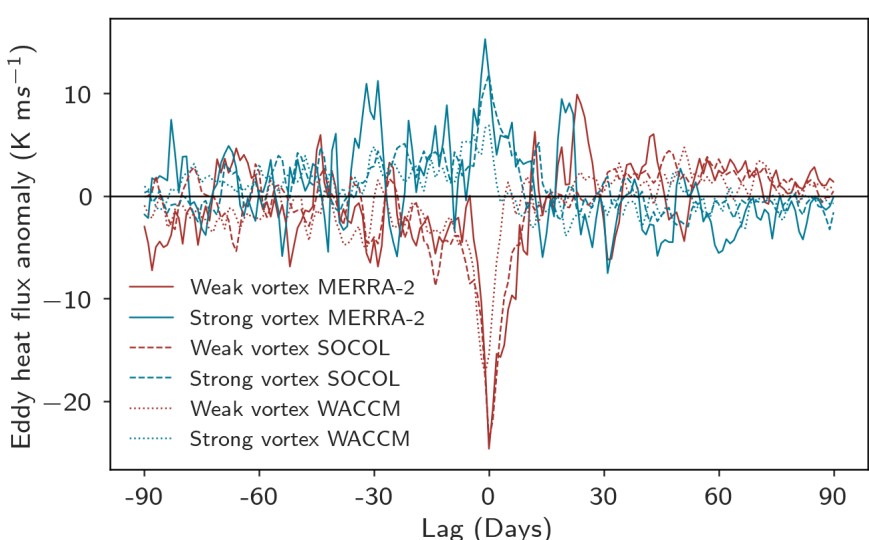

**Figure A3.** Composites of eddy heat flux anomalies (Km/s) averaged over 45-75°S at 100hPa for strong and weak polar vortex events for MERRA-2 and the CCMs SOCOL and WACCM. The central date (lag 0) refers to the onset day of the polar vortex anomaly.

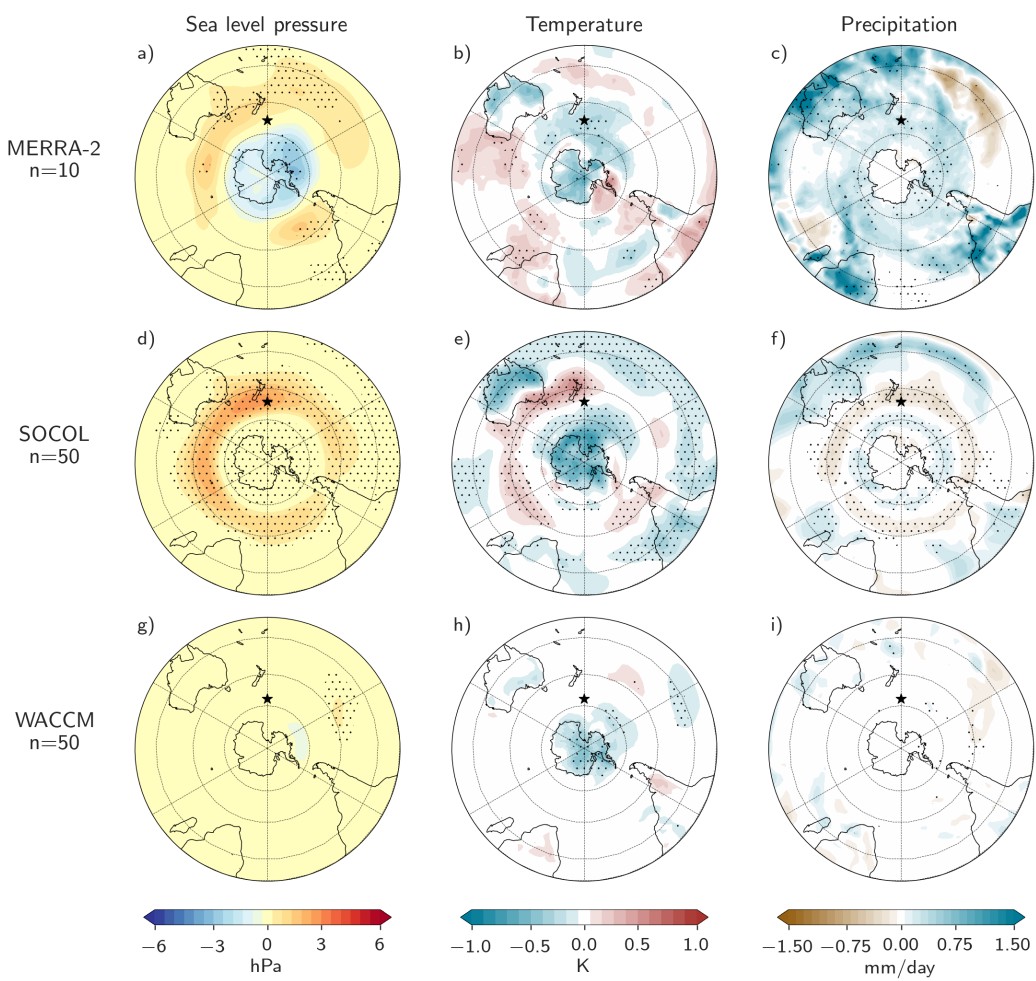

**Figure A4.** Surface climate composites for strong polar vortex regimes of October-January SLP anomalies (a,d,g), 2-meter temperature anomalies (b,e,h) and precipitation anomalies (c,f,i). The reanalysis data MERRA-2 (first row) includes 10 weak vortex regimes, and the CCMs SOCOL (second row), and WACCM (third row) each include 50 weak vortex regimes. Stippling refers to significance at the $4.5\,\%$ level assessed with a bootstrapping test.





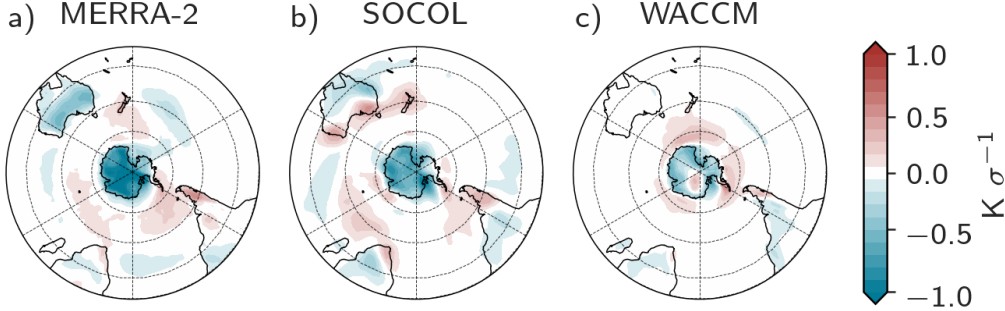

**Figure A5.** Regression of daily 2-meter temperature anomalies on the daily $1000\,\text{hPa}$ SAM index for the October to January time period for the reanalysis data MERRA-2 (a) for the time period 1980-2020 and the CCMs SOCOL (b) and WACCM (c) for the 200 model years each.



*Data availability.* The modeling data used in this study will be made available in the ETH Research Collection. The MERRA2 reanalysis data are available from the Goddard Earth Sciences Data and Information Services Center (GES DIC) https://disc.gsfc.nasa.gov/datasets? project=MERRA-2.

390 *Code and data availability.* All codes and scripts used for the analysis in this study are available from the corresponding author upon reasonable request.

*Author contributions.* G.C. conceived the modelling experiments, G.C. and M.F. conducted the modelling experiments, N.B., G.C. and M.F. processed the data, N.B., G.C., M.F., D.D., D.W. analysed and interpreted the results. N.B. wrote the paper with input from all authors.

*Competing interests.* The authors declare that they have no conflict of interest.

395 *Acknowledgements.* We thank Prof. Dr. Thomas Peter for very useful discussions. Support from the Swiss National Science Foundation through Ambizione Grant PZ00P2_180043 for N.B., G.C and M.F. is gratefully acknowledged. Support from the Swiss National Science Foundation through project PP00P2_198896 to D.D. is gratefully acknowledged.





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
