# Peer review of "Exploring the link between austral stratospheric polar vortex anomalies and surface climate in chemistry-climate models"

_Atmospheric Chemistry and Physics, 2022_

## Author Response (AR1)

**Reply to reviews**

Nora Bergner, Marina Friedel, Daniela I.V. Domeisen, Darryn Waugh, Gabriel Chiodo

August 31, 2022

RC = Reviewer Comment
AR = Author Reply

**Reviewer 1**

**RC 1.1**  The paper aims to explore the robustness of the tropospheric response to major stratospheric 'extreme' events in a reanalysis and two chemistry-climate models 'on a wide range of timescales'. The topic is important and outcomes of such an analysis are potentially valuable for the communities in the Southern Hemisphere. The paper is well structured and clearly written; conclusions are supported by the analysis.

However, I think that the aim of the analysis has not been achieved and, perhaps, could not have been achieved using the selected models. The reason must be that models don't capture the intensity of polar vortex anomalies and tropospheric SAM impacts well enough to asses robustness of the stratospheric-stratospheric coupling and surface impacts. Hence, I think that the title and parts of introduction need to be re-written to be consistent with the skills of the models.

**AR 1.1**  Thank you for this comment. We would like to point out that the bulk of the polar vortex anomalies is represented by the two models and that the magnitude of polar vortex anomalies come reasonably close to the 2019 event, although no event in the magnitude of 2002 occurs in the models (see Fig. 1 in the manuscript). Assessing robustness cannot be achieved using observations alone, and models are the best, albeit imperfect, tools that are currently available. It is also one point of the paper to document the model shortcomings and biases in the stratosphere-troposphere coupling and comparing two different models allows us to draw conclusions despite their biases. Nevertheless, we would agree that using the term 'robustness' could be misinterpreted and we would therefore suggest to adapt the title to: *Exploring the linkage between austral stratospheric polar vortex anomalies and surface climate* and adapt our introduction to: *In this study, we aim to explore linkages between polar vortex anomalies and their surface response in the*

*SH in reanalysis data and chemistry-climate models (CCMs). In particular, we investigate the sensitivity of stratosphere-troposphere coupling and linkages to surface climate to the short observational record, internal variability, and model biases.*

**RC 1.2** Furthermore, I am wondering why all simulations are forced with the same GHGs, ODS, boundary conditions of the year 2000. Byrne et al. (2019) have shown that ENSO affects the eddy-driven jet via the stratosphere, with no evidence of a direct tropospheric pathway. Therefore, how results of the experiments are sensitive to the use of different initial conditions? Moreover, MERRA composites include various ENSO (and other large-scale drivers) phases, hence, how fair is to compare model and reanalysis composites?

**AR 1.2** The primary objective of this study is to investigate the impact of interannual variability of the polar vortex on surface climate. Simulations forced with constant boundary conditions eliminate the effects of climate change and ozone trends due to long-term changes in ODSs, aiding the purpose of this paper. When comparing our time-slice simulations with historical simulations (1980-2020) of the chemistry climate model inter-comparison (CCMI-1) (Morgenstern et al., 2017) using CESM1 WACCM and SOCOL, the stratospheric SAM anomalies are reasonably similar to reanalysis. In fact, the time-slice simulations (referred to as "const. BCs" in Figure 1) get closer to the more extreme events, such as 2019 (Figure 1). For the tropospheric SAM signal in Figure 2, the simulations show slightly different results: the CCMI-1 historical simulations of SOCOL show a weaker signal following weak polar vortex anomalies than our time-slice simulations, whereas the opposite is the case for WACCM. The differences between our runs and CCMI simulations are most likely due to the different set-ups: CCMI-1 runs are much shorter (40 years vs our own 200 years time-slice experiments), they include ozone and climate-change trends, and in the case of the CCMI-1 runs from SOCOL, SSTs are prescribed.

We wish to clarify that the time-slice simulations of both models that we are using are coupled to an ocean model and therefore include various ENSO states. Hence, these experiments also capture any ENSO effects on the tropospheric and stratospheric SAM, and are directly comparable to MERRA-2. Based on the Nino 3.4 index which we show in Figure 3, we see that the stratospheric extremes occur during all ENSO phases, which renders models and reanalysis comparable. There is a weak negative correlation between ENSO and the stratospheric SAM in reanalysis and one model (SOCOL) captures it (Spearman correlation coefficients based on monthly averaged indexes of Nino 3.4 and 10 hPa SAM: MERRA-2: $-0.11$, p=0.01; SOCOL: $-0.16$, p=$3.18 \times 10^{-15}$; WACCM: $-0.01$, p=0.44). While it would be interesting to further investigate the impact of ENSO teleconnections and other initial conditions on the stratosphere, it is outside the aim and the scope of this study.

[Figure]

Figure 1: The 10 hPa SAM index values of the 25 % lowest and highest springtime SAM indices in MERRA-2 (10 events per weak/strong category), and in CCMs WACCM and SOCOL for simulations using constant boundary conditions (50 events per weak/strong category), and the reference "ref-C2" simulations from CCMI-1 (Morgenstern et al., 2017), which are referred to as "hist" (solely focusing on 1980-2020 for comparability with MERRA-2 reanalysis, i.e. 10 events per weak/strong category). The most extreme events in the reanalysis data are annotated with the year of occurrence.

[Figure]

Figure 2: Distribution of the mean 500 hPa SAM index averaged over the October-January period following weak and strong polar vortex anomalies for MERRA-2 (a), and CCMs SOCOL (b), and WACCM (c) with each constant BC simulations and historical simulations (1980-2020). The box extends from the lower to upper quartile of the data, the whiskers extend from the lower quartile −1.5 IQR to the upper quartile +1.5 IQR. Data points outside of the whiskers are shown as circles. The horizontal line marks the median value and the triangle indicates the mean of the distribution, which is annotated next to the box.

[Figure]

Figure 3: The figure shows the evolution of the Nino3.4 index, estimated using the 5-month average of surface temperatures in the Nino3.4 region. The detected polar vortex anomalies are marked with red circles for weak events, and blue circles for strong events.

**RC 1.3** Therefore, while the paper is well written, (1) more work is needed to bring the goals of the research in line with actual skills of SOCOL and WACCM; (2) other experiments may be needed that produce polar vortex anomalies of magnitudes that are similar to the observed values (particularly, for WACCM) and that account for other boundary and initial conditions.

**AR 1.3** We are in favour of the referee's suggestions, but are of the opinion that the suggested additional experiments are outside of the aim and the scope of this study. Additionally, experiments that produce polar vortex anomalies of similar magnitudes to the observed values do come with new challenges and questions, e.g. the nudging timescales, parameters, height range etc, that could determine the effect of the nudging. Nudging has not been systematically evaluated for CCMI models and e.g. Chrysanthou et al. (2019) shows that nudging does not necessarily lead to improvements in the stratospheric residual circulation. Therefore, while the suggested nudged experiments would potentially be beneficial for an improved understanding of linkages between stratosphere and troposphere in our models, further work would be needed to evaluate and test such experiments. Thus, experiments with a nudged stratospheric circulation are interesting material for an extensive follow-up study.

Moreover, we wish to note that while 2002 and 2019 show the strongest surface responses, there is still a surface effect, particularly over Antarctica, when leaving these two most extreme years out (Fig. 4 ). Leaving out 2002 and 2019 makes the polar vortex anomalies comparable between models and observations.

Other comments:

**RC 1.4** By extreme events, the authors mean top/bottom 25% SAM index of all years. Those events may be called anomalous, but not extreme. Probably +/- 1 std would be a better threshold.

**AR 1.4** We agree that using the term "extreme" for the 25% top and bottom years is misleading. We chose this percentile as a trade off between a sufficiently large sample size and events that are still anomalous enough to produce a surface result. Furthermore, using this percentile includes the same events in the reanalysis as in previous studies (Thompson et al., 2005; Byrne and Shepherd, 2018). For the above reasons, we have not adapted the threshold, but we now refer to these events as "anomalous" in the manuscript, instead of "extreme".

**RC 1.5** l. 296, Fig. A5: 'Model biases in the jet position are consistent with the CCMs' tropospheric SAM surface patterns' - what do you mean? I agree that there are biases in the jet location (Fig. 7), but for the temperature response I don't see biases, I see

[Figure]

Figure 4: Surface composites following weak polar vortex anomalies without the 2002 and 2019 events in MERRA-2.

lack of skills outside of polar regions and high midlatitudes. This is particularly true for Australia. This plot illustrates why, as I stated earlier, these models could not be used to study surface impacts of polar vortex events outside of Antarctica. Therefore, I'd suggest limiting the analysis to the processes that are reasonably well simulated by the models (e.g., jets).

**AR 1.5**  We mean that the tropospheric SAM in models projects differently on surface climate (e.g., temperature and precipitation) than in reanalysis, as is visible in Figure A5. Furthermore, these biases in the SAM surface pattern (of the tropospheric SAM in general, not necessarily related to the downward coupling of the stratosphere) are likely related to the model biases. For instance, as the tropospheric jet is typically more equatorward in SOCOL, the corresponding temperature patterns are slightly shifted equatorward as well. To clarify, we therefore reformulate to: *Model biases in the jet position are consistent with the CCMs' tropospheric SAM surface pattern, as a northward shift implies a northward shift of the temperature patterns, and vice versa.*

**RC 1.6**  Fig. 8: For WACCM, instead of saying that 'tropospheric SAM timescales are shorter than in the reanalysis', it would be good to say that the stratospheric SAM does not quite reach the surface. Also, weak stratospheric SAM in WACCM becomes very obvious.

**AR 1.6**  We would like to emphasise that the SAM timescale is a constructed value and does not directly represent the SAM. From Figure 2 in the manuscript, we see that the SAM anomaly in WACCM reaches the surface for intermittent time periods. This means that although the stratospheric SAM reaches the surface in WACCM, the surface SAM response decays rapidly due to the small time scale in the troposphere (i.e. indicating too little persistence of the tropospheric circulation) in WACCM.

**RC 1.7**  Considering the strength of modelled polar vortices, perhaps most extreme modelled vortices can be compared with vortex anomalies in the reanalysis of the same magnitude? Alternatively (or additionally), other experiments can be conducted that amplify the polar vortex anomalies to observed values to assess their impact.

**AR 1.7**  In our bootstrap analysis and Figure 6 in the manuscript, we look at subsampled composites (so the number of samples is the same for the models and reanalysis data) and here we can compare similar magnitudes of polar vortex anomalies to the average surface temperature over Antarctica and Australia:

For Antarctica, when using similar stratospheric SAM magnitudes (red box in Fig. 5), the Antarctic temperature anomaly between models and reanalysis becomes comparable, although models show a larger spread (likely due to the larger number of years available).

[Figure]

Figure 5: Scatterplots of 500 synthetic bootstrapped composites (n=10) for Antarctic (a,c) and Australian (e,g) 2-meter temperature anomalies in weak polar vortex regimes vs. the composite stratospheric 10 hPa SAM peak anomaly. The $r$-value refers to the Pearson correlation coefficient between the two quantities, the line refers to the fitted linear regression with the 95 % confidence interval of the slope in shading. The kernel density estimation (KDE) of the temperature composites is shown on the right of the scatterplots (b,d,f,h), and the mean 2 m temperature values for the 10 weak polar vortex regimes are marked with red dashes for the reanalysis datasets.

For the Australian temperature anomaly, only very few composites are comparable to the reanalysis but the bulk shows lower temperatures. This leads to the conclusion that the lack of extreme stratospheric weak events in models is not responsible for the missing surface response.

**RC 1.8**   l. 331: if the observed warming signal over Antarctica and Australia is robust, then do you need to use models at all?

**AR 1.8**   Even though the bootstrapped analysis suggests that these signals are robust, they are nevertheless constrained to a small number of observations (that can be resampled with replacement). We compared our results to another reanalysis dataset (JRA-55) (Figure 6) and see surprisingly large differences between JRA-55 and MERRA-2, which is likely also influenced by the fact that over polar regions, less data is available to be assimilated and the reanalysis product is more strongly influenced by the model. Climate models aid in the understanding of the underlying processes and causalities. Therefore, it is also relevant to evaluate how they simulate certain processes and document biases that could motivate future improvements.

Minor:

**RC 1.9**   Fig. 3: Three subplots can be merged into one plot, similar to Fig. 1. There is enough space alone X axis to show six box-whiskers (colours clearly separate strong/weak events).

**AR 1.9**   We have adjusted the figure accordingly (Fig. 7).

**RC 1.10**   Fig. A2: please add labels for Y axis

**AR 1.10**   We thank the reviewer for pointing this out. We have added the labels (Fig. 8).

**RC 1.11**   l. 333, 'we find that the Australian temperature signal is even more uncertain than the warming over Antarctica': The fact that Australian signal is more uncertain than warming over Antarctica would be expected, please re-word.

**AR 1.11**   We reformulated to *In the model experiments, we find that the Australian temperature signal is very uncertain, with equal likelihoods of warming and cooling.*

[Figure]

Figure 6: Scatterplots of 500 synthetic bootstrapped composites (n=10) including JRA-55 reanaylisis data for Antarctic (a,c) and Australian (e,g) 2-meter temperature anomalies in weak polar vortex regimes vs. the composite stratospheric 10 hPa SAM peak anomaly. The $r$-value refers to the Pearson correlation coefficient between the two quantities, the line refers to the fitted linear regression with the 95 % confidence interval of the slope in shading. The kernel density estimation (KDE) of the temperature composites is shown on the right of the scatterplots (b,d,f,h), and the mean 2 m temperature values for the 10 weak polar vortex regimes are marked with red dashes for the reanalysis datasets.

[Figure]

Figure 7: Distribution of the mean 500 hPa SAM index averaged over the October-January period following weak and strong polar vortex anomalies for MERRA-2 (a), SOCOL (b), and WACCM (c). The box extends from the lower to upper quartile of the data, the whiskers extend from the lower quartile −1.5 IQR to the upper quartile +1.5 IQR. Data points outside of the whiskers are shown as circles. The horizontal line marks the median value and the triangle indicates the mean of the distribution, which is annotated next to the box.

[Figure]

Figure 8: Ozone standard deviation in austral spring in MERRA-2 (a), SOCOL (b) and WACCM (c).

**Reviewer 2**

This paper investigates the link between Southern Hemisphere stratospheric polar vortex variability and spring-summer surface climate. It compares the representation of this in both reanalysis and two chemistry-climate models (CCMs). The text concludes that there is, in general, a robust relationship between stratospheric extremes and surface climate, but that there are differing biases in the two models, which may be related to their climatological states.

Southern Hemisphere stratosphere-troposphere coupling is a topical issue, highlighted by recent extremes in surface climate such as Australian wildfires, large interannual stratospheric variability, and a background emergence of ozone hole recovery. Understanding the representation of this in climate models, as this paper aims to do, is therefore of great value. I found this paper to be clearly written, logically structured, and with clear figures that support the conclusions. I have a main concern around the choice of model simulations that are analysed, and consequences for the general applicability of the conclusions. I also include some more minor comments below, and I hope that the authors find these helpful.

Major comment:

**RC 2.1**   1. The motivation for using chemistry climate models for this study is unclear to me, particularly given the analysis in appendix A1 showing that interactive chemistry makes relatively little difference in the stratosphere-troposphere coupling. I think that the relative expense of these simulations, meaning the study has just two models, limits the robustness of the results. For instance, it is very difficult to draw any strong conclusions about the relationship presented between model climatologies and stratosphere-troposphere coupling, as in section 3.4, from just two models. I think that the paper would benefit significantly from the inclusion of a much broader range of models, for at least this part of the analysis. For instance, preindustrial control simulations from CMIP6 are readily available and would be suited for this analysis.

**AR 2.1**   The main motivation to use chemistry climate models is that important processes in the stratosphere are better represented (e.g. Eyring et al., 2010) and observational and model studies suggest the importance of ozone on surface climate based on correlations (Bandoro et al., 2014; Gillett et al., 2019). Including interactive chemistry in simulations is also shown to be important for the surface response in individual events (Hendon et al., 2020), on subseasonal timescales (Jucker and Goyal, 2022), and in the Northern Hemisphere (Friedel et al., 2022). A priori, we did not know that the interactive chemistry in these simulations on seasonal timescales makes relatively little difference in the Southern Hemisphere, which is why we have included these findings in the Appendix.

We agree that it is difficult to draw strong conclusions on the relationship between climatology and stratosphere-troposphere coupling from only these two models. For this reason, we have analyzed ref-C2 simulations from CCMI models (Morgenstern et al., 2017) focusing only on the time period 1980-2020. We chose the CCMI simulations over the CMIP6 preindustrial control, as their boundary conditions are more comparable to the reanalysis data, the stratosphere is better resolved, and they include the chemistry interactions as in our simulations. However, they also come with different caveats, as there are different setups between the simulations: some are forced with observed SSTs, while others are coupled to an ocean model and they also include trends as the reanalysis data. In particular, the imposed SSTs may not be consistent with the stratospheric variability and may interfere with tropospheric SAM responses. In that regard, our simulations with constant boundary conditions, a coupled ocean and 200 model years have an advantage for investigating the interannual variability. In spite of these caveats, the results from the CCMI historical simulations (ref-C2) confirm some of the results from the SOCOL and WACCM time-slice simulations.

We summarize the new findings based on CCMI as follows:

- The tropospheric SAM response (500 hPa) in spring-to-summer (October-January) following weak polar vortex events is inversely related to the stratospheric (50 hPa) SAM timescale (Fig 9), as evident from the statistically significant correlation (r=0.71). The persistence of circulation anomalies in the lower stratosphere has been shown to be an important indicator for the surface response of SSWs in the Northern Hemisphere (Runde et al., 2016; Karpechko et al., 2017). Here, we confirm the existence of this relationship in a very different context (stratosphere-troposphere coupling in the SH).

- The average surface temperature response in spring-to-summer (October-January) in Australia and Antarctica is correlated with the stratospheric (50 hPa) SAM timescale (Fig. 10 and Fig. 11).

- The jet latitude and tropospheric SAM timescale are positively correlated, but the relationship between jet latitude and the spring-to-summer (October-January) surface temperature response in Australia is less clear (Fig. 12). The surface response consists of a complex interplay between these different metrics (SAM timescale and jet latitude), and models with an equatorward jet and long SAM timescales tend to have a larger surface response (e.g. CMAM).

Overall, the CCMI model results support our conclusions that WACCM possibly underestimates and SOCOL overestimates the downward stratospheric impact due to their biases in the SAM timescale and jet latitude. Particularly the models' SAM timescales appear to impact the surface response of stratospheric anomalies. The differing surface patterns as a result of different jet latitudes are not directly confirmed by the CCMI results as

some models show a strong warming over Australia despite an even more biased midlatitude jet stream than SOCOL (e.g. CMAM, CCSRNIES-MIROC3.2). However, prescribed vs. modelled SSTs make direct comparison difficult as these likely influence the surface impacts as well. The model simulations that show reasonably similar results to the reanalysis data are also closer to the reanalysis in terms of jet latitude and SAM timescales (e.g.MRI-ESM1r1, NIWA-UKCA).

[Figure]

Figure 9: The mean Oct-Jan 50 hPa SAM timecale in each model versus the mean Oct-Jan 500 hPa SAM response following anomalous stratospheric weak events. The Spearman correlation coefficient and p-value are annotated in the upper right corner.

[Figure]

Figure 10: The mean Oct-Jan 50 hPa SAM timecale in each model versus the mean Antarctic temperature response following anomalous stratospheric weak events. The Spearman correlation coefficient and p-value are annotated in the upper right corner.

[Figure]

Figure 11: The mean Oct-Jan 50 hPa SAM timecale in each model versus the mean Australian temperature response following anomalous stratospheric weak events. The Spearman correlation coefficient and p-value are annotated in the upper right corner.

[Figure]

Figure 12: The mean Oct-Jan tropospheric jet latitude in each model versus the mean Australian temperature response following anomalous stratospheric weak events. The Spearman correlation coefficient and p-value are annotated in the upper right corner.

Minor comments:

**RC 2.2** 2. L7-8: I'd encourage against using this bracket construction. While it saves a small amount of space, it requires the reader to read the sentence twice.

**AR 2.2** We would rephrase to:
*The CCMs show a similar downward propagation of polar vortex anomalies as the reanalysis data: weak polar vortex anomalies are on average followed by a negative tropospheric Southern Annular Mode (SAM) in spring to summer, while strong polar vortex anomalies are on average followed by a positive SAM.*

**RC 2.3** 3. L81: I question whether linear detrending is appropriate in the case of the reanalysis given that we have a nonlinear forcing (ozone depletion and stabilization/recovery). Perhaps the detrending can be split into two time periods to reflect this, or some more evidence presented that linear detrending is acceptable.

**AR 2.3** We agree that in this time period, nonlinear forcings existed (e.g., ozone depletion and recovery). However, the trends are small compared to the interannual variability and comparing e.g. the tropospheric SAM response or the surface composites including / excluding linear detrending yield very similar results, as can be seen in Fig. 13 and Fig. 14. We also note that linear detrending has been used in previous studies on this subject (Lim et al., 2019), so we use this method for better comparability.

[Figure]

Figure 13: As Fig. 3 in the manuscript, but without detrending the data.

[Figure]

Figure 14: MERRA-2 Oct-Jan surface composite following weak polar vortex anomalies with detrended data (a), and with not detrended data (b).

**RC 2.4**   4. L122: A little more detail is needed on how strong/weak polar vortex events are defined. i.e. how are they distinguished from final warmings, and is there a minimum time gap between consecutive events?

**AR 2.4**   In this study, we only choose one event per year, as the timescale in the SH is so long that the westerly flow typically does not recover after a deceleration (i.e., weak anomaly) and only one event per season is typically observed (Thompson et al., 2005; Gerber et al., 2010). We do not directly distinguish weak vortex events from final warmings, since we only select the events based on ranking bottom/top 25% of SAM anomalies in August-November. Hence, early final warmings could in principle be counted as weak events (approximately one time in MERRA-2 and WACCM, 20 times in SOCOL, when we calculate the final warming dates based on the reversal of the zonal mean zonal wind at 10 hPa and 60°S without returning above the threshold for more than 10 consecutive days (Butler and Domeisen, 2021)). Not making the distinction between mid-winter weak vortex events and final warmings can be justified by the fact that early final warmings – especially in the southern hemisphere – are driven by the same mechanism as mid-winter vortex weakenings.

We would adapt this part in the manuscript to make it clearer: *As the SAM variance peaks in austral spring, we detect the largest and smallest anomaly in the daily 10 hPa SAM index between August and November each year. This allows only one weak or strong vortex event per year, which is reasonable as the dynamical timescales in the SH are long enough that the westerly flow remains weak/strong after a perturbation (Gerber et al., 2010). From these values, we define the highest and lowest 25 % as the strong and weak polar vortex events. Therefore, we obtain 10 strong/weak polar vortex events in the reanalysis data and 50 strong/weak events in the CCMs. We do not define a minimum temporal distance to the final stratospheric warming.*

**RC 2.5**   5. L138: Is this interpolation to get jet latitude linear? If so it may introduce some errors, and this may be worth checking against the calculations detailed in the TropD package (Adam et al. 2018, Geosci. Model Dev., doi:10.5194/gmd-11-4339-2018)

**AR 2.5**   The interpolation of the zonal mean wind is based on a spline interpolation of second order (python scipy package). We compared the jet latitude index against the output of the TropD package and got almost identical results for MERRA-2 and WACCM. SOCOL differed a bit, but our jet index appears very reasonable when checking the underlying wind data (interpolated and not interpolated).

**RC 2.6**   6. L149: I think this would benefit from some more detail on how the bootstrapping works so that the reader doesn't have to refer to those references to check this

important point. My guess is that the same calendar dates are used as the stratospheric events (to preserve seasonality) but the year is randomly varied?

**AR 2.6**   The idea of the bootstrapping is to create composites of anomalous stratospheric events with different combinations of tropospheric states, similar to running ensemble simulations in a model. We resample different combinations of the 10 observed events, so by necessity repeating some events and leaving others out (resampling with replacement). We describe this resampling in l. 149-153: "The observed composite consists of 10 events with tropospheric states that are unrelated to the stratospheric signal. We randomly resample the 10 observed events with replacement to form 500 synthetic composites. In the synthetic composites, we allow an individual event to be repeated a maximum of three times. We thereby estimate how much the surface signal varies between the synthetic composites and how it relates to the strength of the polar vortex anomaly."

We realize this description may not have been clear enough for the reader to fully grasp how the method works. We would slightly adapt this in the manuscript, to make it clearer: *We randomly resample the 10 observed events with replacement to form 500 synthetic composites consisting of different combinations of the observed events, by necessity repeating some events and leaving other events out. In the synthetic composites, we allow an individual event to be repeated a maximum of three times.*

**RC 2.7**   7. L175: I'm not sure that saying the anomalies 'propagate down' is necessarily accurate here. For instance, the appearance of anomalies in the stratosphere and troposphere is at almost the same time for reanalysis and SOCOL (i.e. it appears barotropic).

**AR 2.7**   We changed the formulation to: *The anomalies peak in the mid- to upper stratosphere following the onset date (by construction) and persist in the lower stratosphere for up to 90 days (Fig. 2 a,b), consistent with similar previous observational analyses (Thompson et al., 2005; Byrne and Shepherd, 2018).*

**RC 2.8**   8. L204-205: "We primarily focus on weak polar vortex events, for which the observed tropospheric SAM response is larger than for strong polar vortex events". Is the larger tropospheric response to weak vortex events due to the fact that weak vortex events are on average stronger (i.e. larger 10 hPa SAM anomaly) or because the stratosphere-troposphere coupling is stronger following them? I think this would be worthwhile expanding on a little.

**AR 2.8**   This is indeed an interesting point and the asymmetry was first noted in Thompson et al. (2005). While it is likely related to the larger stratospheric anomalies, we do not

know whether the stratosphere-troposphere coupling is different for the same magnitude of anomalies, which could be investigated in future studies.

**RC 2.9**  9. L210: Using the term 'datasets' to refer to model simulations may be a little confusing.

**AR 2.9**  We rephrased to: *However, the magnitude and spatial extent of the SLP signal differs among the reanalysis data and model simulations, with a much weaker signal in WACCM than in SOCOL, consistent with the differences among these models in their tropospheric SAM response (Fig. 3).*

**RC 2.10**  10. L265-269: It is stated that it is unlikely that differences arise from short observational record. I think that this is an important point and suggest that it could this be tested quantitatively through some statistical testing.

**AR 2.10**  The mean of the bootstrapped Australian temperature anomalies in MERRA-2 and SOCOL are significantly different from each other based on a two-sample two-sided t-test, and respectively between MERRA-2 and WACCM. The average of the Australian temperature anomaly in MERRA-2 is 0.17 K. The corresponding quantile in the distribution of the subsampled composites is 0.93 in SOCOL, and 0.86 WACCM. Hence, both models are unlikely ($<33$ %) to capture the observed temperature anomaly. We thank the reviewer for raising this point and would like to add this new quantiative estimate to the manuscript.

**RC 2.11**  11. L288-289: Note that Simpson and Polvani (2016) (cited in the paper) find that the jet latitude-shift relationship does not hold in summer. I think this should be discussed here and perhaps any conflicting conclusions with that study clarified.

**AR 2.11**  Simpson and Polvani (2016) show that the correlation between SH jet shift (with climate change) and jet latitude only holds in the annual mean, but not in summer. In our study, we do not consider the long-term jet shifts due to climate change. Instead, we explore the relationship between interannual variations in the stratospheric polar vortex and the tropospheric SAM in spring-to-summer. Hence, our results are not directly comparable to Simpson and Polvani (2016).

**References**

Bandoro, J., S. Solomon, A. Donohoe, D. W. J. Thompson, and B. D. Santer (Aug. 2014). "Influences of the Antarctic Ozone Hole on Southern Hemispheric Summer Climate Change". In: *Journal of Climate* 27.16, pp. 6245–6264. DOI: 10.1175/JCLI-D-13-00698.1.

Butler, A. H. and D. I. V. Domeisen (May 2021). "The wave geometry of final stratospheric warming events". In: *Weather and Climate Dynamics* 2.2, pp. 453–474. DOI: 10.5194/wcd-2-453-2021.

Byrne, N. J. and T. G. Shepherd (May 2018). "Seasonal Persistence of Circulation Anomalies in the Southern Hemisphere Stratosphere and Its Implications for the Troposphere". In: *Journal of Climate* 31.9, pp. 3467–3483. DOI: 10.1175/JCLI-D-17-0557.1.

Chrysanthou, A., A. C. Maycock, M. P. Chipperfield, S. Dhomse, H. Garny, D. Kinnison, H. Akiyoshi, M. Deushi, R. R. Garcia, P. Jöckel, O. Kirner, G. Pitari, D. A. Plummer, L. Revell, E. Rozanov, A. Stenke, T. Y. Tanaka, D. Visioni, and Y. Yamashita (2019). "The effect of atmospheric nudging on the stratospheric residual circulation in chemistry–climate models". In: *Atmospheric Chemistry and Physics* 19.17, pp. 11559–11586. DOI: 10.5194/acp-19-11559-2019.

Eyring, V., T. G. Shepherd, and D. W. Waugh (2010). *SPARC CCMVal Report on the Evaluation of Chemistry-Climate Models*. Tech. rep. Backup Publisher: SPARC Publication Title: SPARC Report Volume: No. 5. SPARC Office, 426 pp.

Friedel, M., G. Chiodo, A. Stenke, D. I. V. Domeisen, S. Fueglistaler, J. G. Anet, and T. Peter (July 2022). "Springtime arctic ozone depletion forces northern hemisphere climate anomalies". In: *Nature Geoscience* 15.7, pp. 541–547. DOI: 10.1038/s41561-022-00974-7.

Gerber, E. P., M. P. Baldwin, H. Akiyoshi, J. Austin, S. Bekki, P. Braesicke, N. Butchart, M. Chipperfield, M. Dameris, S. Dhomse, S. M. Frith, R. R. Garcia, H. Garny, A. Gettelman, S. C. Hardiman, A. Karpechko, M. Marchand, O. Morgenstern, J. E. Nielsen, S. Pawson, T. Peter, D. A. Plummer, J. A. Pyle, E. Rozanov, J. F. Scinocca, T. G. Shepherd, and D. Smale (2010). "Stratosphere-troposphere coupling and annular mode variability in chemistry-climate models". In: *Journal of Geophysical Research: Atmospheres* 115.D3. DOI: https://doi.org/10.1029/2009JD013770.

Gillett, Z. E., J. M. Arblaster, A. J. Dittus, M. Deushi, P. Jöckel, D. E. Kinnison, O. Morgenstern, D. A. Plummer, L. E. Revell, E. Rozanov, R. Schofield, A. Stenke, K. A. Stone, and S. Tilmes (June 2019). "Evaluating the Relationship between Interannual Variations in the Antarctic Ozone Hole and Southern Hemisphere Surface Climate in Chemistry–Climate Models". In: *Journal of Climate* 32.11, pp. 3131–3151. DOI: 10.1175/JCLI-D-18-0273.1.

Hendon, H. H., E.-P. Lim, and S. Abhik (Aug. 2020). "Impact of Interannual Ozone Variations on the Downward Coupling of the 2002 Southern Hemisphere Stratospheric

Warming". In: *Journal of Geophysical Research: Atmospheres* 125.16. DOI: 10.1029/2020JD032952.

Jucker, M. and R. Goyal (2022). "Ozone-Forced Southern Annular Mode During Antarctic Stratospheric Warming Events". In: *Geophysical Research Letters* 49.4, e2021GL095270. DOI: https://doi.org/10.1029/2021GL095270.

Karpechko, A. Y., P. Hitchcock, D. H. W. Peters, and A. Schneidereit (2017). "Predictability of downward propagation of major sudden stratospheric warmings". In: *Quarterly Journal of the Royal Meteorological Society* 143.704, pp. 1459–1470. DOI: 10.1002/qj.3017.

Lim, E.-P., H. H. Hendon, G. Boschat, D. Hudson, D. W. J. Thompson, A. J. Dowdy, and J. M. Arblaster (Nov. 2019). "Australian hot and dry extremes induced by weakenings of the stratospheric polar vortex". In: *Nature Geoscience* 12.11, pp. 896–901. DOI: 10.1038/s41561-019-0456-x.

Morgenstern, O., M. I. Hegglin, E. Rozanov, F. M. O'Connor, N. L. Abraham, H. Akiyoshi, A. T. Archibald, S. Bekki, N. Butchart, M. P. Chipperfield, M. Deushi, S. S. Dhomse, R. R. Garcia, S. C. Hardiman, L. W. Horowitz, P. Jöckel, B. Josse, D. Kinnison, M. Lin, E. Mancini, M. E. Manyin, M. Marchand, V. Marécal, M. Michou, L. D. Oman, G. Pitari, D. A. Plummer, L. E. Revell, D. Saint-Martin, R. Schofield, A. Stenke, K. Stone, K. Sudo, T. Y. Tanaka, S. Tilmes, Y. Yamashita, K. Yoshida, and G. Zeng (Feb. 2017). "Review of the global models used within phase 1 of the Chemistry–Climate Model Initiative (CCMI)". In: *Geoscientific Model Development* 10.2, pp. 639–671. DOI: 10.5194/gmd-10-639-2017.

Runde, T., M. Dameris, H. Garny, and D. E. Kinnison (2016). "Classification of stratospheric extreme events according to their downward propagation to the troposphere". In: *Geophysical Research Letters* 43.12, pp. 6665–6672. DOI: https://doi.org/10.1002/2016GL069569.

Simpson, I. R. and L. M. Polvani (2016). "Revisiting the relationship between jet position, forced response, and annular mode variability in the southern midlatitudes". In: *Geophysical Research Letters* 43.6, pp. 2896–2903. DOI: 10.1002/2016GL067989.

Thompson, D. W. J., M. P. Baldwin, and S. Solomon (Mar. 2005). "Stratosphere–Troposphere Coupling in the Southern Hemisphere". In: *Journal of Atmospheric Sciences* 62.3, pp. 708–715. DOI: 10.1175/JAS-3321.1.